# Hypervirulent *Klebsiella pneumoniae* Causing Neonatal Bloodstream Infections: Emergence of NDM-1-Producing Hypervirulent ST11-K2 and ST15-K54 Strains Possessing pLVPK-Associated Markers

Subhankar Mukherjee,[a]* Punyasloke Bhadury,[b] Shravani Mitra,[a] (ID) Sharmi Naha,[a] Bijan Saha,[c] Shanta Dutta,[a] (ID) Sulagna Basu[a]

[a]Division of Bacteriology, ICMR-National Institute of Cholera and Enteric Diseases, Kolkata, West Bengal, India
[b]Integrative Taxonomy and Microbial Ecology Research Group, Department of Biological Sciences, Indian Institute of Science Education and Research Kolkata, Mohanpur, Nadia, West Bengal, India
[c]Department of Neonatology, Institute of Post-Graduate Medical Education & Research and SSKM Hospital, Kolkata, West Bengal, India

**ABSTRACT** *Klebsiella pneumoniae* is a major cause of neonatal sepsis. Hypervirulent *Klebsiella pneumoniae* (hvKP) that cause invasive infections and/or carbapenem-resistant hvKP (CR-hvKP) limit therapeutic options. Such strains causing neonatal sepsis have rarely been studied. Characterization of neonatal septicemic hvKP/CR-hvKP strains in terms of resistance and virulence was carried out. Antibiotic susceptibility, molecular characterization, evaluation of clonality, *in vitro* virulence, and transmissibility of carbapenemase genes were evaluated. Whole-genome sequencing (WGS) and mouse lethality assays were performed on strains harboring pLVPK-associated markers. About one-fourth (26%, 28/107) of the studied strains, leading to mortality in 39% (11/28) of the infected neonates, were categorized as hvKP. hvKP-K2 was the prevalent pathotype (64.2%, 18/28), but K54 and K57 were also identified. Most strains were clonally diverse belonging to 12 sequence types, of which ST14 was most common. Majority of hvKPs possessed virulence determinants, strong biofilm-forming, and high serum resistance ability. Nine hvKPs were carbapenem-resistant, harboring $bla_{NDM-1}$/$bla_{NDM-5}$ on conjugative plasmids of different replicon types. Two NDM-1-producing high-risk clones, ST11 and ST15, had pLVPK-associated markers (*rmpA*, *rmpA2*, *iroBCDEN*, *iucABCDiutA*, and *peg-344*), of which one co-transferred the markers along with $bla_{NDM-1}$. The 2 strains revealed high inter-genomic resemblance with the other hvKP reference genomes, and were lethal in mouse model. To the best of our knowledge, this study is the first to report on the NDM-1-producing hvKP ST11-K2 and ST15-K54 strains causing fatal neonatal sepsis. The presence of pLVPK-associated markers and $bla_{NDM-1}$ in high-risk clones, and the co-transmission of these genes via conjugation calls for surveillance of these strains.

**IMPORTANCE** *Klebsiella pneumoniae* is a leading cause of sepsis in newborns and adults. Among the 2 major pathotypes of *K. pneumoniae*, classical (cKP) and hypervirulent (hvKP), hvKP causes community-acquired severe fatal invasive infections in even healthy individuals, as it possesses several virulence factors. The lack of comprehensive studies on neonatal septicemic hvKPs prompted this work. Nearly 26% diverse hvKP strains were recovered possessing several resistance and virulence determinants. The majority of them exhibited strong biofilm-forming and high serum resistance ability. Nine of these strains were also carbapenem (last-resort antibiotic)-resistant, of which 2 high-risk clones (ST11-K2 and ST15-K54) harbored markers (pLVPK) noted for their virulence, and were lethal in the mouse model. Genome-level characterization of the high-risk clones showed resemblance with the other hvKP reference genomes. The presence of transmissible carbapenem-resistant gene, $bla_{NDM}$, along with pLVPK-markers calls for vigilance, as most clinical microbiology laboratories do not test for them.

Address correspondence to Sulagna Basu, supabasu@yahoo.co.in, or basus.niced@gov.in.

*Present address: Subhankar Mukherjee, Department of Zoology, Government General Degree College, Singur, Hooghly, West Bengal, India.

The authors declare no conflict of interest.

**KEYWORDS** *Klebsiella pneumoniae*, neonatal sepsis, antibiotic resistance, carbapenem resistance, hypervirulence, India

*K*lebsiella pneumoniae is an opportunistic nosocomial pathogen, capable of causing several infecting syndromes in both adults and neonates. It is also the primary cause of neonatal sepsis in lower-middle-income countries (1, 2). Moreover, the global dissemination of carbapenem-resistant *K. pneumoniae* (CRKP) has created a substantial public-health concern (3). Among the carbapenemases, the New Delhi metallo-$\beta$-lactamase gene ($bla_{NDM}$) is the most notorious due to the wide-spectrum of antimicrobial-hydrolyzing ability, presence in diverse conjugative plasmids and sequence types (STs), and the ability to cause large hospital-mediated outbreaks (4–7). Of the 43 NDM-variants that have been identified so far (https://www.ncbi.nlm.nih.gov/pathogens/refgene/#blaNDM), NDM-1 is the most widely disseminated, and exhibits significant resistance to clinically relevant $\beta$-lactams (4). Thus, these strains have been designated as an urgent threat to human health (8, 9).

To date, two major pathotypes of *K. pneumoniae* have been detected in clinical contexts: classical *K. pneumoniae* (cKP) and hypervirulent *K. pneumoniae* (hvKP). In contrast to the highly antimicrobial-resistant (AMR) cKP populations that primarily cause hospital-acquired infections, hvKP, the recently emerging pathotype can cause various community-acquired invasive fatal infections in immunocompetent individuals (10). Unlike cKP, hvKP can effectively sequester iron, often harbor plasmid-borne virulence factors (*rmpA*, *rmpA2*, *iro*, and *iuc*), and are frequently associated with several capsular types (K1, K2, K5, K20, K54, and K57) (11, 12). Though most hvKP strains have thick capsular polysaccharide and are hypermucoviscous, it should be noted that hypervirulence attributes of *K. pneumoniae* may not always be associated with hypermucoviscosity (13).

Since the emergence of hvKP almost 4 decades ago, most reported strains belonged to antibiotic-sensitive populations. However, a changing scenario of antibiotic-resistant hvKP has now been observed in clinical *K. pneumoniae* after the initial description of hvKP. Carbapenem-resistant hvKP (CR-hvKP) pathotypes are now frequently reported, and the clinical landscape of hvKP is altering dramatically (12, 14).

The occurrence of CR-hvKP in clinical settings has driven several recent studies (15, 16), as CR-hvKP pathotypes have the potential to be the 'next generation super-bug' (10). Although characterization of hvKP strains causing infections in adult population has been investigated, a detailed characterization of neonatal septicemic hvKP/CR-hvKP strains in terms of resistance and virulence is still lacking. The vulnerability of this population with limitations in therapeutic options, poses a greater challenge for clinicians. With increasing numbers of carbapenem-resistant neonatal infections, particularly where the causative organism harbors the $bla_{NDM}$, an understanding of the burden of disease due to CR-hvKP is crucial in the neonatal population. Thus, hvKP/CR-hvKP strains recovered from septicemic neonates were investigated in terms of resistance, virulence, and transmission of the $bla_{NDM}$ gene. This study also included genome-level characterization of selected CR-hvKP strains.

## RESULTS

**Antimicrobial susceptibility testing.** Among the recovered 107 *K. pneumoniae*, >50% of strains were resistant to 9 different antimicrobials (Fig. S1). Twenty-nine hvKP strains were detected, where 28 of them are characterized here, as previous work on CR-hvKP strain EN5275 (ST23-K1) has already been published (17).

Most of the recovered hvKPs were resistant to cephalosporin, cephamycin, monobactam, aminoglycosides, fluoroquinolone, and trimethoprim-sulfamethoxazole (Table 1). Among the studied hvKPs, only 32.1% (9/28) strains were detected as metallo-$\beta$-lactamase producers, resistant to carbapenems (MICs ranged from 8 to 64 mg/L), and were categorized as CR-hvKP.

**TABLE 1** Microbiological features of recovered neonatal septicemic carbapenem-susceptible and carbapenem-resistant hvKP strains

| Variable | $n$ (%)[b] | | | |
| | Total, $n$ = 28 (100%) | CS-hvKP, $n$ = 19 (67.8%) | CR-hvKP, $n$ = 9 (32.1%) | $P$ value |
|---|---|---|---|---|
| AMR phenotypes (no. of strains resistant to particular antibiotics) | | | | |
| Piperacillin | 26 (92.8) | 17 (89.4) | 9 (100) | 1.000 |
| Cefotaxime | 23 (82.1) | 14 (73.6) | 9 (100) | 0.769 |
| Cefoxitin | 13 (46.4) | 4 (21) | 9 (100) | 0.042[a] |
| Aztreonam | 21 (75) | 12 (63.1) | 9 (100) | 0.552 |
| Meropenem | 9 (32.1) | 0 (0.0) | 9 (100) | <0.001[a] |
| Amikacin | 15 (53.5) | 6 (31.5) | 9 (100) | 0.108 |
| Gentamicin | 14 (50) | 7 (36.8) | 7 (77.7) | 0.322 |
| Ciprofloxacin | 23 (82.1) | 14 (73.6) | 9 (100) | 0.769 |
| Trimethoprim/Sulfamethoxazole | 23 (82.1) | 14 (73.6) | 9 (100) | 0.769 |
| AMR genotypes | | | | |
| $\beta$-lactams | | | | |
| $bla_{TEM}$ | 16 (57.1) | 9 (47.3) | 7 (77.7) | 0.552 |
| $bla_{SHV}$ | 21 (75) | 13 (68.4) | 8 (88.8) | 0.765 |
| $bla_{OXA}$ | 14 (50) | 7 (36.8) | 7 (77.7) | 0.322 |
| $bla_{CTX-M-15}$ | 20 (71.4) | 11 (57.8) | 9 (100) | 0.384 |
| $bla_{AmpC}$ | 8 (28.5) | 3 (15.7) | 5 (55.5) | 0.216 |
| $bla_{NDM-1}$ | 7 (25) | 0 (0.0) | 7 (77.7) | 0.001[a] |
| $bla_{NDM-5}$ | 2 (7.1) | 0 (0.0) | 2 (22.2) | 0.126 |
| $bla_{OXA-232}$ | 1 (3.5) | 0 (0.0) | 1 (11.1) | 0.344 |
| Aminoglycosides | | | | |
| armA | 2 (7.1) | 0 (0.0) | 2 (22.2) | 0.126 |
| rmtB | 1 (3.5) | 0 (0.0) | 1 (11.1) | 0.344 |
| rmtC | 3 (10.7) | 0 (0.0) | 3 (33.3) | 0.048[a] |
| aac(6')-ib | 12 (42.8) | 3 (15.7) | 9 (100) | 0.017[a] |
| Quinolone resistance genes | | | | |
| qnrB1 | 16 (57.1) | 10 (52.6) | 6 (66.6) | 0.750 |
| qnrS1 | 9 (32.1) | 2 (10.5) | 7 (77.7) | 0.023[a] |
| oqxA | 21 (75) | 12 (63.1) | 9 (100) | 0.552 |
| oqxB | 21 (75) | 12 (63.1) | 9 (100) | 0.552 |
| aac(6')-ib-cr | 17 (60.7) | 10 (52.6) | 7 (77.7) | 0.748 |
| Capsular types | | | | |
| K2 | 18 (64.2) | 12 (63.1) | 6 (66.6) | 1.000 |
| K20 | 1 (3.5) | 0 (0.0) | 1 (11.1) | 0.344 |
| K54 | 5 (17.8) | 3 (15.7) | 2 (22.2) | 1.000 |
| K57 | 4 (14.3) | 4 (21) | 0 (0.0) | 0.303 |
| Virulence factors | | | | |
| Hypermucoviscosity-related gene | | | | |
| rmpA | 2 (7.1) | 0 (0.0) | 2 (22.2) | 0.126 |
| rmpA2 | 2 (7.1) | 0 (0.0) | 2 (22.2) | 0.126 |
| CPS synthesis | | | | |
| wcaJ | 9 (32.1) | 0 (0.0) | 9 (100) | <0.001[a] |
| Lipopolysaccharide synthesis | | | | |
| wabG | 26 (92.8) | 17 (89.4) | 9 (100) | 1.000 |
| uge | 22 (78.5) | 13 (68.4) | 9 (100) | 0.564 |
| Adhesin | | | | |
| fimH | 22 (78.5) | 13 (68.4) | 9 (100) | 0.564 |
| mrkD | 23 (82.1) | 14 (73.6) | 9 (100) | 0.769 |
| Iron uptake and transport | | | | |
| entB | 22 (78.5) | 13 (68.4) | 9 (100) | 0.564 |
| ybtS | 11 (39.2) | 2 (10.5) | 9 (100) | 0.010[a] |
| kfuBC | 25 (89.2) | 16 (84.2) | 9 (100) | 0.780 |
| iucA | 2 (7.1) | 0 (0.0) | 2 (22.2) | 0.126 |
| iutA | 2 (7.1) | 0 (0.0) | 2 (22.2) | 0.126 |
| iroN | 2 (7.1) | 0 (0.0) | 2 (22.2) | 0.126 |

**TABLE 1** (Continued)

| Variable | Total, $n = 28$ (100%) | CS-hvKP, $n = 19$ (67.8%) | CR-hvKP, $n = 9$ (32.1%) | P value |
|---|---|---|---|---|
| Allantoin regulon | | | | |
| *allS* | 8 (28.5) | 4 (21) | 4 (44.4) | 0.421 |
| pLVPK-associated markers | | | | |
| *rmpA*, *rmpA2*, *iroN*, *iucA*, and *iutA* | 2 (7.1) | 0 (0.0) | 2 (22.2) | 0.126 |

[a]P values that denote statistical significance.
[b]CS-hvKP, carbapenem-susceptible hvKP; CR-hvKP, carbapenem-resistant hvKP.

**AMR determinants.** Various $\beta$-lactamase genes (ESBLs, AmpCs, and carbapenemases), aminoglycoside resistance genes, and quinolone resistance genes were identified in hvKPs (Table 1).

Moreover, out of 9 CR-hvKPs, 7 harbored $bla_{NDM-1}$, and 2 harbored $bla_{NDM-5}$. One strain co-harbored $bla_{NDM-5}$ and $bla_{OXA-232}$. Among the several AMR determinants, *rmtC* ($P = 0.048$), *aac(6′)-ib* ($P = 0.017$), and *qnrS1* ($P = 0.023$) were significantly higher in CR-hvKPs (Table 1). Figure 1 depicts the distribution of AMR determinants.

**Virulence-associated analysis.** K2 (64.2%, 18/28) was the most prevalent of the identified capsular types, followed by K54 (17.8%, 5/28) and K57 (14.3%, 4/28) (Table 1). Moreover, K2 and K54 were detected in both carbapenem-resistant and carbapenem-susceptible strains. Several virulence determinants, viz., *wabG*, *kfuBC*, *mrkD*, *fimH*, *entB*, *uge*, *ybtS*, and *wcaJ* were identified in hvKPs; of them, *wcaJ* ($P < 0.001$) and *ybtS* ($P = 0.010$) were significantly higher in CR-hvKPs (Table 1). In contrast, pLVPK-associated markers (*rmpA*, *rmpA2*, *iroN*, *iucA*, and *iutA*) were detected only in 2 NDM-1-producing CR-hvKP strains (EN5180 and EN5289). Figure 1 depicts the distribution of various virulence factors.

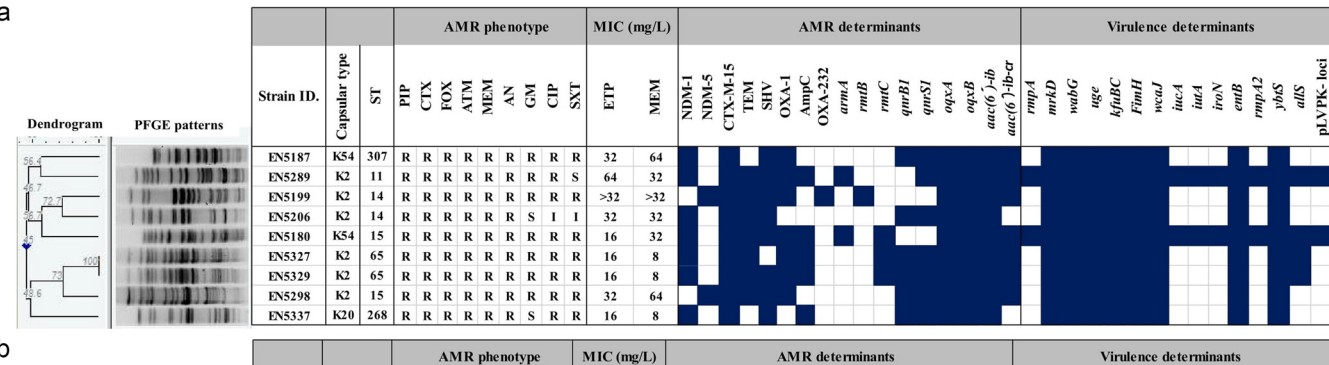

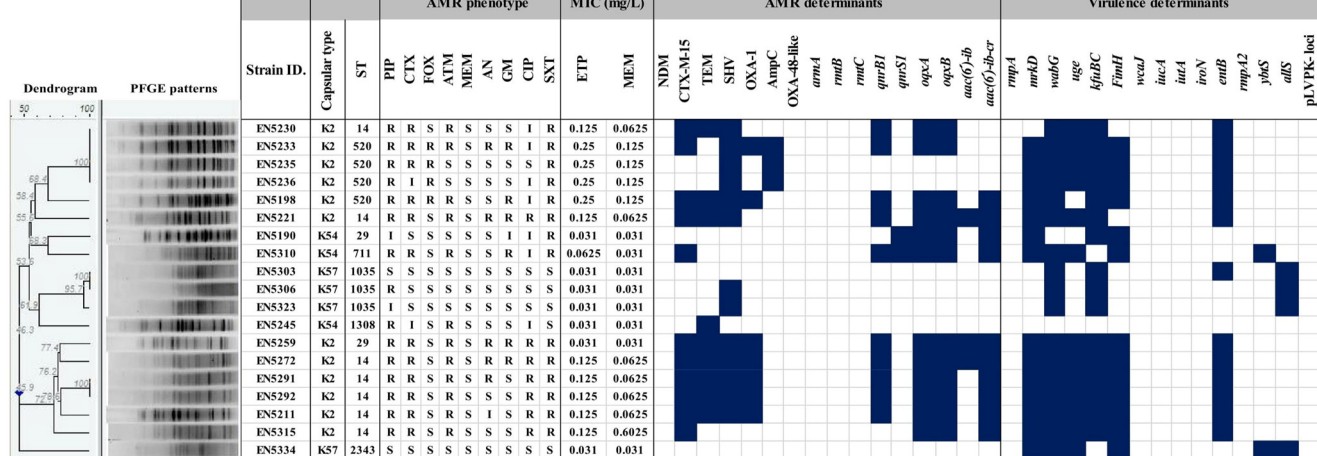

**FIG 1** PFGE profiles along with distribution of different antimicrobial resistance (both phenotypic and genotypic) and virulence determinants of the neonatal septicemic, carbapenem-resistant (a) and carbapenem-susceptible (b) hvKP strains. Dark blue filled-up boxes indicate the presence of different antimicrobial resistance and virulence determinants.

Additionally, 42.8% (12/28) of strains possessed strong biofilm-forming ability ($OD_{595}$ = 0.82 to 1.23) compared to the negative control ($OD_{595}$ = 0.27), and 78.5% (22/28) of strains exhibited high level of serum resistance (100% viable counts after 3 h of incubation) compared to the positive control. Table S1 summarizes *in vitro* virulence results.

**Clonal relatedness among hvKP strains.** Pulsed-field gel electrophoresis (PFGE) revealed that most of the studied hvKPs (60.7%, 17/28) were clonally diverse. However, among the 9 CR-hvKPs, EN5327 and EN5329 were clonal, and among the 19 carbapenem-susceptible strains, 3 distinct clonal-clusters were observed (Cluster I – EN5230, EN5233, EN5235, and EN5236; Cluster II – EN5303, EN5306, and EN5323; Cluster III – EN5291 and EN5292), while the rest were distinct (Fig. 1).

The hvKPs consisted of 12 different STs (ST11, ST14, ST15, ST29, ST65, ST268, ST307, ST520, ST711, ST1035, ST1308, and ST2343) (Fig. 1). Of them, ST14 was the most common (32.1%, 9/28), and was associated with both carbapenem-resistant and carbapenem-susceptible populations. Additionally, ST11, ST15, ST65, ST268, and ST307 were associated only with CR-hvKPs. Other STs (ST29, ST520, ST711, ST1035, ST1308, and ST2343) were detected only in carbapenem-susceptible strains.

**Clinical outcome of the neonates infected with hvKP strains.** Table 2 represents the clinical status of the hvKP-infected neonates. Out of 28 neonates, 16 (57.1%) were outborn (refers to the neonates referred from some other facility or extramural births), and most were delivered preterm (53.5%, 15/28), were of low birth weight (57.1%, 16/28), or extremely low birth weight (17.8%, 5/28). 39.2% (11/28) of neonates who succumbed to the condition, suffered from severe clinical complications, as listed in Table 2. Notably, 33.3% (6/18) of neonates infected with hvKP-K2 strains did not survive.

**Analysis of transconjugants.** Conjugal transfer of $bla_{NDM}$ was successful in all cases. All transconjugants exhibited MIC for carbapenems ranging between 2 and 32 mg/L. However, Tc-EN5199 with dual carbapenemases ($bla_{NDM-5}$ and $bla_{OXA-232}$) showed higher MIC for ertapenem and meropenem (Table 3). $bla_{NDM-1}$/$bla_{NDM-5}$, along with several other AMR determinants, was borne on different conjugative plasmid scaffolds, *viz.*, IncA/C, IncR, IncFII, IncFIIK, and IncHIB-M/FIB-M (Table 3).

Moreover, among the identified transconjugants, only Tc-EN5289 tested positive for $bla_{NDM-1}$, $bla_{CTX-M-15}$, pLVPK-associated markers, and harbored a conjugative IncHIB-M/FIB-M plasmid. This co-transfer of pLVPK-markers along with $bla_{NDM-1}$ suggests the possible association of both highly transmissible AMR and hypervirulence determinants in the same conjugative plasmid (Table 3).

Except for a difference in a few AMR and virulence determinants, no other significant difference was observed between the carbapenem-resistant and carbapenem-susceptible *K. pneumoniae* in terms of *in vitro* virulence assessment and demographics of the neonates.

**Resistome, virulome, and comparative genomic analysis of CR-hvKP EN5180 and EN5289 strains.** The genomes of EN5180 ($\sim$ 6.1 Mb, 56.43% G+C, 5974 CDS) and EN5289 (5.9 Mb, 56.68% G+C, 5902 CDS) harbored a plethora of different AMR determinants, several plasmid replicons, heavy-metal resistance determinants, and efflux pumps and regulator elements (Table 4).

Moreover, EN5180 and EN5289 harbored a diverse array of putative virulence factors, including gene clusters for type 1 and type 3 fimbriae, ABC iron-transporter, aerobactin, enterobactin, salmochelin, yersiniabactin, allantoin utilization, capsule regulators, and multiple type-VI secretion systems (T6SS I - III) (Table 4). WGS confirmed the presence of several pLVPK-associated markers (*rmpA*, *rmpA2*, *iroBCDEN*, *iucABCDiutA*, and *peg-344*) in both the strains (Table 4), which were missing in the other hvKP strains. An alignment of EN5180 and EN5289 with pLVPK (GenBank accession AY378100) demonstrated the existence of pLVPK-like sequence in the studied genomes (Fig. S2). Furthermore, EN5180 and EN5289 possessed several prophage elements (Table S2 and Fig. S3) and CRISPR arrays (Fig. S4).

Comparative genomic (BLAST+ and BLASTN) analysis of EN5180 and EN5289 with the other hvKP reference genomes revealed high inter-genomic resemblance (*in silico* DNA-DNA hybridization values $\sim$ 94%) (Fig. 2 and Table S3). Phylogenomic analysis also showed that EN5180 and EN5289 were highly similar to the other hvKP genomes (Fig. S5).

**TABLE 2** Clinical and demographic characteristics of the septicemic neonates (n = 28) infected with the hvKP and/or CR-hvKP strains

| Strain No. | Capsular type(s) | ST[a](s) | Time of isolation | Clinical complications | Sex | Inborn[b]/outborn[c] | Birth wt (g) | Mode of delivery | Gestational age (wk) | Outcome |
|---|---|---|---|---|---|---|---|---|---|---|
| **Neonates infected with carbapenem-resistant hvKP (n = 9)** | | | | | | | | | | |
| EN5180 | K54 | 15 | April, 2014 | Sepsis, perinatal asphyxia, respiratory distress | M[d] | Inborn | 1,400 | LUCS[f] | 34 | Expired |
| EN5187 | K54 | 307 | June, 2014 | Hyperglycemia, apnea, sepsis, and meningitis | M | Outborn | 995 | NVD[g] | 31 | Expired |
| EN5199 | K2 | 14 | November, 2014 | Gastroschisis | F[e] | Inborn | 2,223 | NVD | 36 | Expired |
| EN5206 | K2 | 14 | February, 2015 | Septic arthritis, perinatal asphyxia, HIE-II[h], intracranial hemorrhage | M | Outborn | 2,250 | NVD | 37 | Expired |
| EN5289 | K2 | 11 | March, 2016 | Agonal breathing, ELBW[i], IVH-III[j], septic shock | M | Outborn | 650 | NVD | 30 | Expired |
| EN5298 | K2 | 15 | April, 2016 | Respiratory distress, ELBW, sepsis, apnea | F | Outborn | 835 | NVD | 28 | Discharged |
| EN5327 | K2 | 65 | June, 2016 | Bilious vomiting, intestinal obstruction | M | Outborn | 1,640 | NVD | 34 | Expired |
| EN5329 | K2 | 65 | July, 2016 | Hyperbilirubinemia | F | Inborn | 2,420 | NVD | 35 | Discharged |
| EN5337 | K20 | 268 | October, 2016 | Perinatal asphyxia, HIE-II | M | Outborn | 2,700 | NVD | 37 | Discharged |
| **Neonates infected with carbapenem-susceptible hvKP (n = 19)** | | | | | | | | | | |
| EN5190 | K54 | 29 | August, 2014 | Born with MSL[k], hematuria | F | Inborn | 2,118 | LUCS | 37 | Discharged |
| EN5198 | K2 | 520 | November, 2014 | Respiratory distress, BCA[l] | F | Outborn | 2,400 | LUCS | 37 | Expired |
| EN5211 | K2 | 14 | April, 2015 | IUGR[m], hyperbilirubinemia, intestinal perforation, culture-positive sepsis, septic shock, DIC[n] | F | Inborn | 850 | NVD | 35 | Expired |
| EN5221 | K2 | 14 | August, 2015 | LBW[o], sepsis | F | Outborn | 1,900 | NVD | 37 | Discharged |
| EN5230 | K2 | 14 | September, 2015 | Perinatal asphyxia, HIE-II | M | Inborn | 3,876 | NVD | 41 | Discharged |
| EN5233 | K2 | 520 | September, 2015 | Respiratory distress syndrome | M | Inborn | 2,164 | LUCS | 35 | Discharged |
| EN5235 | K2 | 520 | September, 2015 | Rh hemolytic disease, hyperbilirubinemia | M | Inborn | 2,784 | LUCS | 38 | Discharged |
| EN5236 | K2 | 520 | September, 2015 | PR[p] Bleeding | F | Inborn | 2,807 | LUCS | 37 | Discharged |
| EN5245 | K54 | 1308 | October, 2015 | Hyperbilirubinemia | M | Outborn | 1,971 | LUCS | 36 | Discharged |
| EN5259 | K2 | 29 | November, 2015 | Poor feeding | M | Outborn | 3,006 | NVD | 37 | Discharged |
| EN5272 | K2 | 14 | December, 2015 | Preterm[q] | F | Outborn | 2,260 | NVD | 27 | Discharged |
| EN5291 | K2 | 14 | March, 2016 | Delayed passage of meconium, Hirschsprung's disease | M | Outborn | 1,900 | NVD | 35 | Discharged |
| EN5292 | K2 | 14 | March, 2016 | Sepsis, DIC, GDM[r] | F | Outborn | 1,550 | LUCS | 37 | Discharged |
| EN5303 | K57 | 1035 | April, 2016 | ELBW, pulmonary hemorrhage | M | Outborn | 945 | NVD | 29 | Expired |
| EN5306 | K57 | 1035 | April, 2016 | Jitteriness, hypoglycemia, hypocalcemia, septic shock | F | Inborn | 2,124 | NVD | 36 | Expired |
| EN5310 | K54 | 711 | May, 2016 | Abdominal distension, intestinal obstruction | M | Outborn | 2,250 | NVD | 38 | Expired |
| EN5315 | K2 | 14 | May, 2016 | Abdominal distension | M | Outborn | 3,074 | NVD | 40 | Discharged |
| EN5323 | K57 | 1035 | June, 2016 | Respiratory distress | M | Outborn | 2,700 | LUCS | 39 | Discharged |
| EN5334 | K57 | 2343 | September, 2016 | Respiratory distress, esophageal atresia, cleft lip and palate | F | Inborn | 1,803 | LUCS | 36 | Discharged |

aST, sequence type.
bInborn, refers to the neonates born at the tertiary care hospital (intramural birth).
cOutborn, refers to the neonates referred from some other facility (extramural birth).
dM, male.
eF, female.
fLUCS, lower uterine segment cesarean section.
gNVD, normal vaginal delivery.
hHIE-II, hypoxic-ischemic encephalopathy grade II.
iELBW, extremely-low-birth weight (<1,000 g).
jIVH-III, intraventricular hemorrhage grade III.
kMSL, meconium-stained liquor.
lBCA, bilateral choanal atresia.
mIUGR, intrauterine growth restriction.
nDIC, disseminated intravascular coagulation.
oLBW, low birth weight (<2,500 g).
pPR bleeding, per rectal bleeding.
qPreterm, <37 weeks of gestational age.
rGDM, gestational diabetes mellitus.

**TABLE 3** Microbiological characteristics of the $bla_{NDM}$-harboring parental strains and their transconjugants

| Strains | ST[a] | MIC (mg/L) | | AMR determinants | pLVPK[d]-associated markers | Inc group |
|---|---|---|---|---|---|---|
| | | ETP[b] | MEM[c] | | | |
| EN5180[f] | 15 | 16 | 32 | $bla_{NDM-1}$, $bla_{CTX-M-15}$, $bla_{TEM-1}$, $bla_{SHV}$, $bla_{OXA}$, $bla_{CMY-6}$, armA, rmtC, oqxAB, aac(6')-ib, aac(6')-ib-cr | rmpA, rmpA2, iucA, iutA, iroN | A/C, FIIK, HIB-M, FIB-M |
| Tc[e]-EN5180[f] | -[g] | 4 | 8 | $bla_{NDM-1}$, $bla_{TEM-1}$, rmtC, aac(6')-ib | - | A/C |
| EN5187 | 307 | 32 | 64 | $bla_{NDM-1}$, $bla_{CTX-M-15}$, $bla_{SHV}$, $bla_{OXA}$, qnrB1, qnrS1, oqxAB, aac(6')-ib, aac(6')-ib-cr | - | FIIK, L/M, FII |
| Tc-EN5187 | - | 8 | 8 | $bla_{NDM-1}$, $bla_{CTX-M-15}$, qnrB1, qnrS1, aac(6')-ib, aac(6')-ib-cr | - | FIIK |
| EN5199 | 14 | >32 | >32 | $bla_{NDM-5}$, $bla_{OXA-232}$, $bla_{CTX-M-15}$, $bla_{TEM-1}$, $bla_{SHV}$, $bla_{OXA}$, rmtB, oqxAB, aac(6')-ib, aac(6')-ib-cr | - | R, FIIK, FII, FIB, FIA, Col |
| Tc-EN5199 | - | 32 | 24 | $bla_{NDM-5}$, $bla_{OXA-232}$, $bla_{CTX-M-15}$, $bla_{TEM-1}$, rmtB | - | R, FIIK, Col |
| EN5206 | 14 | 32 | 32 | $bla_{NDM-1}$, $bla_{CTX-M-15}$, $bla_{TEM-1}$, $bla_{SHV}$, qnrB1, qnrS1, oqxAB, aac(6')-ib | - | FII, FIIK, R |
| Tc-EN5206 | - | 8 | 4 | $bla_{NDM-1}$, $bla_{CTX-M-15}$, qnrS1, aac(6')-ib | - | FIIK |
| EN5289[f] | 11 | 64 | 32 | $bla_{NDM-1}$, $bla_{CTX-M-15}$, $bla_{TEM-1}$, $bla_{SHV}$, $bla_{OXA}$, $bla_{DHA-1}$, armA, qnrS1, oqxAB, aac(6')-ib, aac(6')-ib-cr | rmpA, rmpA2, iucA, iutA, iroN | FIIK, HIB-M, FIB-M, HI1 |
| Tc-EN5289[f] | - | 8 | 4 | $bla_{NDM-1}$, $bla_{CTX-M-15}$ | rmpA, rmpA2, iucA, iutA, iroN | HIB-M, FIB-M |
| EN5298 | 15 | 32 | 64 | $bla_{NDM-5}$, $bla_{CTX-M-15}$, $bla_{TEM-1}$, $bla_{SHV}$, $bla_{OXA}$, qnrB1, qnrS1, oqxAB, aac(6')-ib, aac(6')-ib-cr | - | FIIK, FIB, FIA |
| Tc-EN5298 | - | 4 | 2 | $bla_{NDM-5}$, $bla_{CTX-M-15}$, qnrS1, aac(6')-ib, aac(6')-ib-cr | - | FIIK |
| EN5327 | 65 | 16 | 8 | $bla_{NDM-1}$, $bla_{CTX-M-15}$, $bla_{TEM-1}$, $bla_{OXA}$, $bla_{ACT-69}$, rmtC, qnrB1, qnrS1, oqxAB, aac(6')-ib, aac(6')-ib-cr | - | FII, FIIK, X3 |
| Tc-EN5327 | - | 8 | 4 | $bla_{NDM-1}$, $bla_{CTX-M-15}$, $bla_{TEM-1}$, rmtC, qnrB1, aac(6')-ib, aac(6')-ib-cr | - | FII |
| EN5329 | 65 | 16 | 8 | $bla_{NDM-1}$, $bla_{CTX-M-15}$, $bla_{TEM-1}$, $bla_{SHV}$, $bla_{OXA}$, $bla_{ACT-69}$, rmtC, qnrB1, qnrS1, oqxAB, aac(6')-ib, aac(6')-ib-cr | - | FII, FIIK, X3 |
| Tc-EN5329 | - | 8 | 4 | $bla_{NDM-1}$, $bla_{CTX-M-15}$, $bla_{TEM-1}$, rmtC, qnrB1, aac(6')-ib, aac(6')-ib-cr | - | FII |
| EN5337 | 268 | 16 | 8 | $bla_{NDM-1}$, $bla_{CTX-M-15}$, $bla_{SHV}$, $bla_{DHA-1}$, qnrB1, qnrS1, oqxAB, aac(6')-ib | - | R, FIIK, HIB-M, FIB-M |
| Tc-EN5337 | - | 4 | 2 | $bla_{NDM-1}$, $bla_{CTX-M-15}$, qnrB1, qnrS1, aac(6')-ib | - | R, FIIK |

[a]ST, sequence type.
[b]ETP, ertapenem.
[c]MEM, meropenem.
[d]pLVPK, large virulence plasmid of *K. pneumoniae*.
[e]Tc, *E. coli* J53 transconjugants selected in LB agar medium containing 1 mg/L ertapenem and 100 mg/L sodium azide.
[f]pLVPK-markers-harboring CR-hvKP strains and their respective Tc.
[g]-, absence of any particular attribute.

In addition, *in silico* virulence plasmid (pLVPK-like) analysis of EN5289 revealed that $bla_{NDM-1}$ co-existed with several pLVPK-markers (*rmpA2, iroB, iroN, iucABCDiutA*, and *peg-344*) on the same ~ 210 kb IncHI1B/IncFIB plasmid, henceforth designated as pvirEN5289 (Fig. 3a). $bla_{NDM-1}$ was associated with IS26 and $bla_{CTX-M-15}$ with ISEcp1 (at their upstream). Notably, the presence of type IV secretion system (T4SS) and the conjugal transfer (*tra*) genes in pvirEN5289, firmly substantiates its conjugative nature. The result of the aforementioned analysis in combination with the transconjugant ([Tc]-EN5289) data (Table 3), strongly suggests that EN5289 harbored a conjugative $bla_{NDM-1}$, and hypervirulence determinants co-encoding hybrid plasmid, pvirEN5289. Additionally, another ~ 220 kb hybrid IncHI1B/IncFIB plasmid was detected in EN5180, designated as pvirEN5180 (Fig. 3b), which also co-harbored $bla_{CTX-M-15}$ and various pLVPK-markers (*iroB, iroN, iucABiutA*, and *peg-344*). Unlike EN5289, pLVPK-markers and $bla_{NDM-1}$ were carried on separate plasmids in EN5180, as $bla_{NDM-1}$ was transferred via conjugation but no pLVPK-markers were detected in Tc-EN5180.

Notably, sequence alignments with BLAST revealed that pvirEN5180 and pvirEN5289 shared ~ 82 and ~ 81% query coverage with pLVPK (GenBank accession AY378100), respectively.

**TABLE 4** Genome-level characteristics of neonatal septicemic CR-hvKP strains EN5180 and EN5289

| Variable | Strain information | |
|---|---|---|
| Strain name | *K. pneumoniae* EN5180 | *K. pneumoniae* EN5289 |
| Sequence type | ST15 | ST11 |
| Capsular type | K54 | K2 |
| **Genomic features** | | |
| Genome size (bp) | 6,131,075 | 5,967,685 |
| Contig no. | 115 | 107 |
| GC content (%) | 56.43 | 56.68 |
| CDS[a] | 5,974 | 5,902 |
| rRNA genes | 14 | 34 |
| tRNA genes | 74 | 80 |
| tmRNA[b] genes | 1 | 1 |
| ncRNA[c] | 13 | 11 |
| Pseudogenes | 146 | 135 |
| Repeat region | 3 | 1 |
| **Antibiotic resistance genes** | | |
| Aminoglycoside resistance | *aac(6′)-Ib3, aadA1, armA, rmtC* | *aac(6′)-Ib, aadA1, armA* |
| β-lactam resistance | $bla_{SHV-28}$, $bla_{SHV-106}$, $bla_{TEM-1A}$, $bla_{OXA-9}$, $bla_{CTX-M-15}$, $bla_{CMY-6}$, $bla_{NDM-1}$ | $bla_{SHV-182}$, $bla_{TEM-1B}$, $bla_{OXA-9}$, $bla_{CTX-M-15}$, $bla_{DHA-1}$, $bla_{LAP-2}$, $bla_{NDM-1}$ |
| Quinolone resistance | *aac(6′)-Ib-cr, oqxA, oqxB* | *aac(6′)-Ib-cr, oqxA, oqxB, qnrS1* |
| Fosfomycin resistance | *fosA* | *fosA* |
| Macrolide resistance | *msr*(E) | *msr*(E) |
| Sulphonamide resistance | *sul1* | *sul1* |
| Trimethoprim resistance | –[f] | *tet*(A) |
| **Plasmid types** | | |
| Col plasmids | Col440I | Col440I, Col(BS512) |
| F plasmids | IncFIB(pKPHS1), IncFIB(pNDM-Mar), IncFIB(pQil), IncFII(K) | IncFIB(pNDM-Mar), IncFIB(pQil), IncFII(K) |
| Other plasmids | IncHI1B(pNDM-MAR), IncA/C2 | IncHI1B(pNDM-MAR) |
| **Heavy metal resistance genes** | | |
| Silver resistance | *silCERS* | *silCERS* |
| Tellurite resistance | *terABCDEWXYZ* | *terABCDEWXYZ* |
| **Efflux pumps and regulators** | | |
| RND[d] family efflux pump | *acrAB* | *acrAB, mexB, mdtL* |
| HTH[e] type transcriptional regulator | *acrR* | *acrR* |
| AraC/XylS family transcriptional activator | *marAB, soxRS, rob, ramA, rarA* | *marAB, soxRS, rob, ramA, rarA* |
| TetR family transcriptional regulator | *ramR* | *ramR* |
| HTH LuxR type transcriptional regulator | *sdiA* | *sdiA* |
| rRNA transcriptional activator | *fis* | *fis* |
| Suppressor of efflux pump | *envR* | *envR* |
| Quinolone and Olaquindox efflux pump | *oqxABR* | *oqxABR* |
| **Virulence genes** | | |
| Type 3 fimbriae | *mrkABCDFHIJ* | *mrkABCDFHIJ* |
| Type 1 fimbriae | *fimABCDEFGHIK* | *fimABCDEFGHIK* |
| Type IV pili biosynthesis | - | *pilU* |
| ABC iron transporter | *kfuABC* | *kfuABC* |
| Aerobactin | *iucABCD*; *iutA* | *iucABCD*; *iutA* |
| Enterobactin | *entABCDEFS*; *fepABCDG*; *fes* | *entABCDEFS*; *fepABCDG*; *fes* |
| Salmochelin | *iroBCDEN* | *iroBCDEN* |
| Yersiniabactin | *fyuA*; *irp1, irp2*; *ybtAEPQSTUX* | *fyuA*; *irp1, irp2*; *ybtAEPQSTUX* |
| Allantoin utilization | *allABCDRS* | *allABCDRS* |
| Capsule regulators | *rcsAB*; *rmpA, rmpA2* | *rcsAB*; *rmpA, rmpA2* |
| pLVPK-derived loci | *rmpA, rmpA2*; *iroBCDEN*; *iucABCDiutA*; *peg-344* | *rmpA, rmpA2*; *iroBCDEN*; *iucABCDiutA*; *peg-344* |
| Secretion system | T6SS (I–III) | T6SS (I–III) |

**TABLE 4** (Continued)

| Variable | Strain information | |
| --- | --- | --- |
| Other factors (Autotransporter, Iron/manganese uptake, and Stress adaptation) | *flu*; *sitABCD*; *mntB* | - |
| BioProject no. | PRJNA684006 | |
| Accession no. | JAELUV000000000 | JAELUW000000000 |

*a*CDS, coding sequence.
*b*tmRNA, transfer-mRNA.
*c*ncRNA, non-coding RNA.
*d*RND, resistance-nodulation-division.
*e*HTH, helix-turn-helix.
*f*-, absence of any particular attribute.

*In vivo* **virulence of EN5180 and EN5289.** The Kaplan-Meier survival analysis revealed that mice infected with EN5289 portrayed virulence similar to that of the hvKP-K1 positive control strain SB42 (Fig. 4), indicating that EN5289 was equivalently virulent to the well-established hvKP-K1 strain in this model. However, despite harboring all the hypervirulent genetic traits, higher inoculum ($10^5$ CFU) of EN5180 did not lead to significant mouse lethality.

## DISCUSSION

The recent emergence of antibiotic-resistant hvKP is a serious public-health concern (15). Due to the absence of awareness about hvKP and the lack of a universally agreed consensus definition/marker for hypervirulence, clinical microbiology laboratories are unable to differentiate hvKP from cKP in routine diagnosis, potentially affecting the treatment regimen (18, 19).

Several recent studies have shown that *K. pneumoniae* is the major cause of neonatal sepsis (2); however, such studies lack a separate investigation regarding hvKPs.

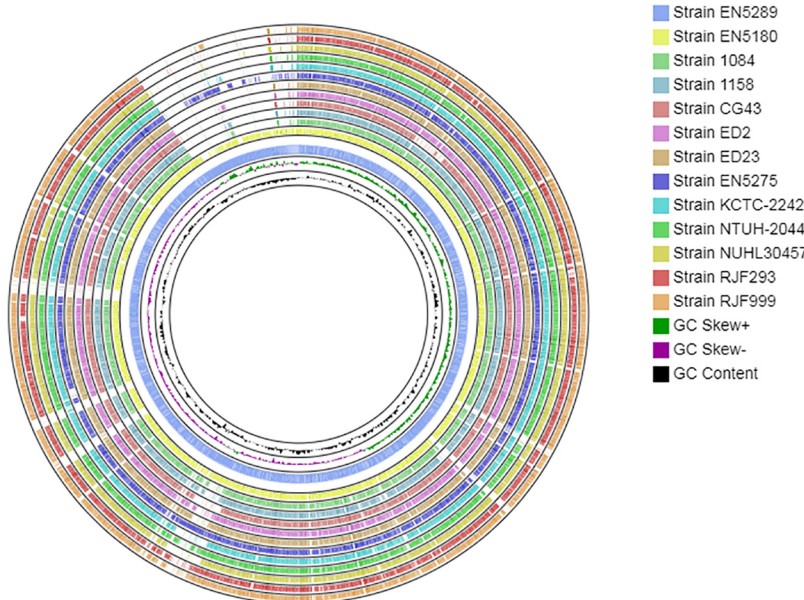

**FIG 2** Comparative BLASTN analysis of NDM-1-producing neonatal septicemic CR-hvKP strains EN5180 and EN5289 and 11 other hvKP reference strains. The circular map was generated using the CGView^BETA comparison tool. From the center to the periphery: Ring 1 and 2 show the GC content and GC skew, respectively; Ring 3 and ring 4 display CR-hvKP strains EN5289 (accession: JAELUW000000000) and EN5180 (accession: JAELUV000000000), respectively; Ring 4: strain 1084 (accession: CP003785.1); Ring 5: strain 1158 (accession: CP006722.1); Ring 6: strain CG43 (accession: CP006648.1); Ring 7: strain ED2 (accession: CP016813.1); Ring 8: strain ED23 (accession: CP016814.1); Ring 9: strain EN5275 (accession: VINI00000000); Ring 10: strain KCTC-2242 (accession: CP002910.1); Ring 11: strain NTUH-2044 (accession: NC_012731.1); Ring 12: strain NUHL30457 (accession: CP026586.1); Ring 13: strain RJF293 (accession: CP014008.1); Ring 14: strain RJF999 (accession: CP014010.1). The sequence comparison revealed that the 13 hvKP strains were shown to have a high degree of similarity.

a

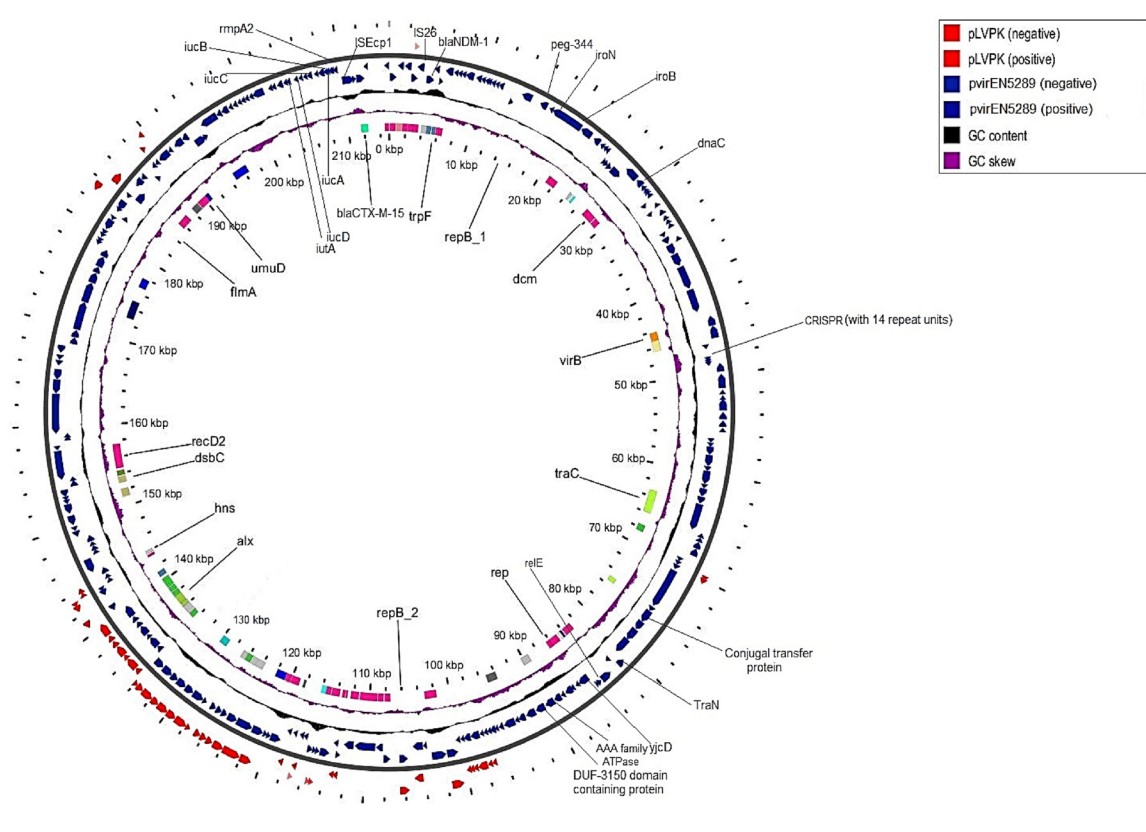

b

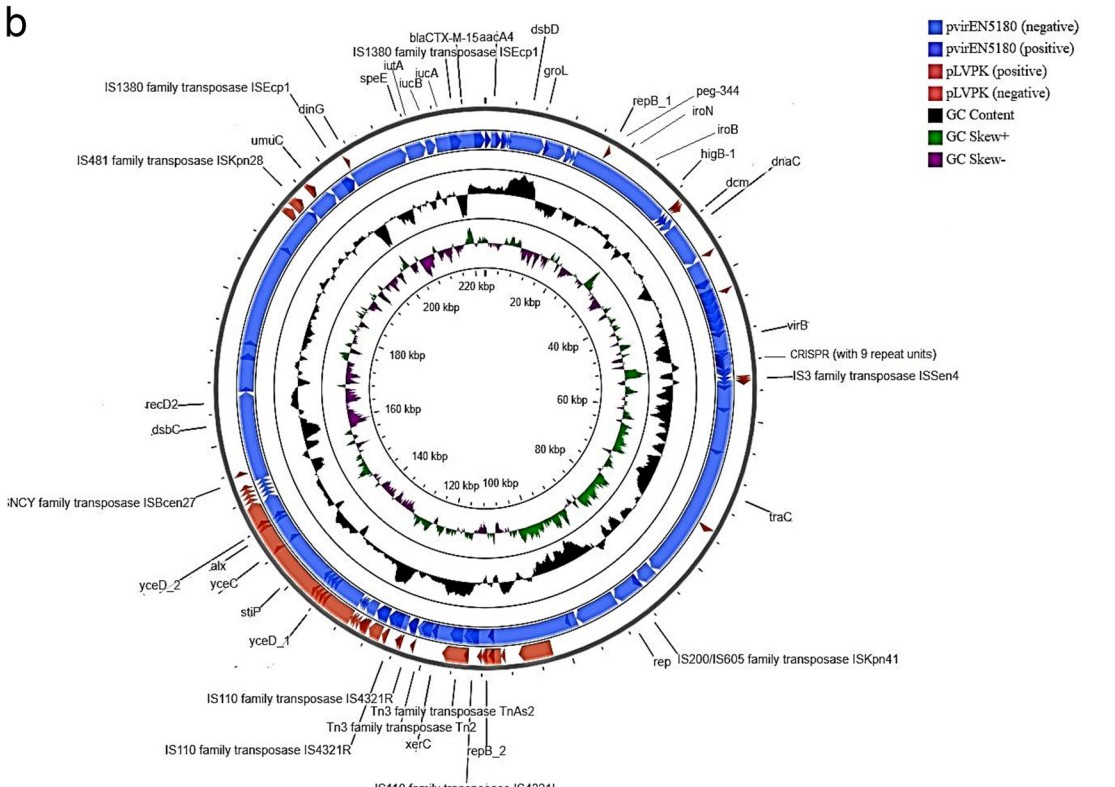

**FIG 3** Virulence plasmid sequence alignment analysis among the $bla_{NDM-1}$ and hypervirulence determinants co-encoding hybrid plasmid, pvirEN5289, and the previously reported virulence plasmid, pLVPK (GenBank accession AY378100) (a), sequence alignment

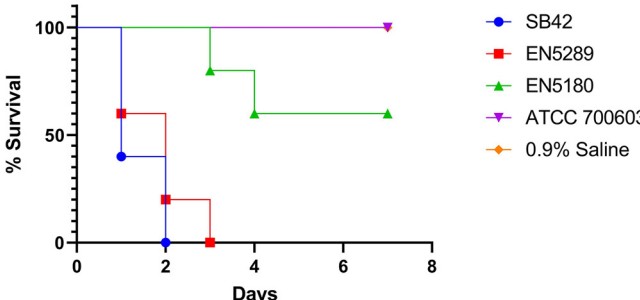

**FIG 4** Kaplan–Meier survival curves for neonatal septicemic, carbapenem-resistant hypervirulent *K. pneumoniae* EN5180- and EN5289-infected mice. Female BALB/c mice (4 to 6-weeks-old, 5 mice/ strain) were intraperitoneally challenged with $1 \times 10^5$ CFU of the different *K. pneumoniae* strains (EN5180, EN5289, hypervirulent control strain SB42, and low virulence control strain ATCC 700603). Saline was administered as vehicle control. Mice were monitored every 24 h up to 7 days, and scored for death. During the 7-day period, no mice in the ATCC 700603 or saline groups died.

Reports of neonatal hvKP cases are just a handful (19), and there are no studies where hvKP strains causing neonatal sepsis have been characterized in terms of resistance and virulence over an extended period of time. To understand hvKP in a new and vulnerable epidemiological context, this study was carried out.

To date, there is no consensus on the definition of hvKP. Early studies have differentiated hvKP by the severity of clinical representation or hypermucoviscous phenotype, and later by capsular/sequence types or presence of certain plasmid-borne virulence markers (14). A consensus is required to interpret characteristics (cKP or hvKP) and compare strains globally (18). The characterization of neonatal strains here endeavors to fill some gaps in this knowledge. Considering the immune-evasive and severe invasive disease-causing ability of some capsular types (K1, K2, K5, K20, K54, and K57) (14), hvKP strains in this study were selected based on the above-stated capsular types.

Nearly 26% of the neonatal septicemic strains were categorized as hvKP based on the capsular types. As there are no reports of prevalence of hvKP in neonatal population, we compared our result with the hvKP cases causing infections in adult population. Our findings revealed that the incidence of hvKP was lower than those reported from China (33%), Taiwan (38%), and Korea (42.4%) (20–22), but higher than the cases reported from Spain (5.4%) and Canada (8.2%) (23, 24). This differential prevalence of hvKP might be partly attributed to the geographical and/or demographical differences, and the characteristics selected to define hvKP strains.

Majority of the studied hvKPs (64.2%, 18/28) belonged to capsular type K2, which also resulted in 33.3% (6/18) of neonatal mortality. The occurrence of K2 in hvKP cases were reported from China (42.9% to 68.7%) (25, 26). However, a large proportion of studies also isolated K1 strains from China and Taiwan, but no K1 serotype was identified from this setup, except one septicemic CR-hvKP ST23 (OXA-232-producing) strain that has been reported earlier (17).

The studied hvKP strains were diverse, belonging to 12 different STs, some of which (ST11, ST14, ST15, ST29, ST65, ST268) have recently been detected in several other hvKP cases (15, 16, 27). The diversity of clones in the study could be from different exposures, as most hvKP-infected neonates were referred from other hospitals with exposure to different nosocomial and/or environmental factors. Previous studies have noted that hvKP-K1 strains belong predominantly to clonal cluster 23 (CC23), and can be grouped into a distinct monophyletic clade. In contrast, hvKP-K2 strains belong to genetically unrelated groups. The diversity of the studied strains is consistent with

**FIG 3** Legend (Continued)

analysis among the $bla_{CTX-M-15}$ and hypervirulent markers co-harboring hybrid plasmid, pvirEN5180 and pLVPK (b). Although, both the genomes harbored all the hypervirulent biomarkers, some of the molecular markers were not portrayed in the given virulence plasmid maps due to their low sequence identity against the available sequence databases.

these reports (28). Such convergence of hvKP in diverse STs indicates a plausible clonal expansion of the pathotype, which is worrisome.

In this study, carbapenem resistance in CR-hvKPs was primarily conferred by $bla_{NDM}$. One strain with dual carbapenemases ($bla_{NDM-5}$ and $bla_{OXA-232}$) was also detected. The presence of $bla_{NDM}$, on a variety of conjugative plasmids, suggests the promiscuous nature of this gene (5). To date, most of the reported CR-hvKPs were found to be associated with $bla_{OXA-48-like}$ and/or $bla_{KPC-2}$ (15–17); however, our study, in concordance with the other recent reports, demonstrated that the NDM-1-producing *K. pneumoniae* are also gradually becoming hypervirulent (29, 30), making the clinical scenario complicated.

hvKP strains often feature several pLVPK-borne markers (*rmpA*, *rmpA2*, *iro*, *iuc*, and/or *peg-344*) (12), which are essential for the hypervirulent phenotype. Of the 28 hvKPs, only 2 NDM-1-producing strains (EN5180 and EN5289) possessed such markers. Apart from harboring diverse AMR determinants, EN5180 and EN5289 were equipped with a wide range of virulence/hypervirulence determinants, and exhibited strong biofilm-forming and high serum resistance abilities. We speculate that these determinants probably enabled them to persist in the antibiotic-laden nosocomial environments, and also elicit colonization within the hosts (31). Although EN5180- and EN5289-infected neonates succumbed to the condition, the mouse lethality assay carried out in this study suggested that only EN5289 was equivalently virulent to the hvKP-K1 strain SB42. The severity of illness (very low birth weight, prematurity, and several clinical complications) in the neonate infected with EN5180, could be a reason for the fatal outcome. The presence of hypervirulent markers (*rmpA*, *rmpA2*, *iroBCDEN*, *iucABCDiutA*, and *peg-344*), pLVPK-like sequence, pLVPK-associated replicon type (IncHI1B/IncFIB), high inter-genomic resemblance with the other hvKP genomes, both *in vitro* and *in vivo* virulence potential, and fatal clinical outcome confirmed the hypervirulence of the EN5180 ST15-K54 and EN5289 ST11-K2 strains.

Most hvKP strains were, until recently, susceptible to various antimicrobials. Strains, such as ST11 and ST15, which are well-known international high-risk CRKP clones causing several hospital-mediated outbreaks, previously lacked the hypervirulence genes. However, studies suggested penetration of virulence plasmids into these clones (15). To date, there are no reports of hvKP/CR-hvKP ST11 and ST15 strains causing neonatal sepsis. Hence, to the best of our knowledge, this study is the first to describe a detailed microbiological, molecular, and genome-level characterization of the NDM-1-producing CR-hvKP ST11-K2 and ST15-K54 strains causing fatal neonatal bloodstream infections. Notably, EN5180 was recovered in 2014, implying that NDM-1-producing CR-hvKP strains might have emerged even before the initial description of $bla_{KPC-2}$-harboring CR-hvKP strains, reflecting the importance of this study (32).

Moreover, as noted in recent studies (33), a new form of molecular convergence is also evident in the studied EN5289 ST11-K2 strain harboring a conjugative $bla_{NDM-1}$ and hypervirulence determinants co-encoding hybrid plasmid. Co-transmission of such newly-emerged plasmids could directly convert a cKP into CR-hvKP, leading to escalation in prevalence of CR-hvKPs in clinical settings. Recently, in a review, we discussed how CRKP limits the already available treatment options for neonates (19), where the occurrence of CR-hvKP compounds the problem.

Given the looming threat of hvKP/CR-hvKP and the severity of disease associated with it, several preclinical studies have recently suggested various potential strategies to combat the hvKP infections either via the development of capsular polysaccharide-based vaccine and/or anti-capsular monoclonal antibody-based immunotherapy (34). The epidemiological landscape of hvKP is, thus, important for the implementation of these therapeutic approaches. Moreover, apart from the therapeutic perspective, a consensus about the definition, the detection of hvKP by a cost-effective method, understanding the evolution of CR-hvKP, and the possible modes of acquisition of such strains in the neonates need attention. Our study provides a glimpse of the prevalence of hvKPs in a given neonatal setup. Further, multi-centric surveillance studies combining clinical and microbiological features, are needed to better understand the transmission mechanisms and to prevent these strains from further propagating in hospital settings.

## MATERIALS AND METHODS

**Ethics.** This study was approved by the Institutional Ethics Committee of the ICMR-National Institute of Cholera and Enteric Diseases (ICMR-NICED) (No. A-1/2016-IEC). Patient data were anonymized and deidentified prior to analysis. Animal experiments were approved by the Institutional Animal Ethical Committee of the ICMR-NICED (No. NICED/CPCSEA/68/GO/(25/294)/2019-IAEC/SB/1).

**Bacterial strains.** During 2014 to 2016, among a total of 285 culture-positive cases, 107 (2014, $n = 19$; 2015, $n = 42$; 2016, $n = 46$) non-duplicate *K. pneumoniae* were recovered from blood of septicemic neonates admitted to the level III unit (Neonatal intensive care unit) of an Indian tertiary care hospital. Blood specimens from the sick neonates were collected at the hospital as part of the routine diagnostic procedure. As a standard protocol, 1 mL of blood for culture was drawn with strict aseptic precautions from a peripheral vein of neonates suspected with sepsis, and inoculated in BD Bactec Peds Plus Vial (Becton Dickinson, MD, USA). Blood culture was performed with the automated Bactec 9050 system (Becton, Dickinson). Gram staining was performed for any culture that flagged positive, and the subculture was carried out on appropriate media based on Gram's stain: MacConkey agar and 5% sheep blood agar (Difco Laboratories, Detroit, MI, USA) for Gram-negative and Gram-positive isolates, respectively. Bacterial identification and antibiotic susceptibility were performed by Vitek-2 compact system and disk diffusion (as per CLSI guidelines) (35), respectively.

**Experimental procedures.** All 107 *K. pneumoniae* were screened for capsular types, *viz.*, K1, K2, K5, K20, K54, and K57, considered as frequently encountered hvKP genotypes (36, 37). For the capsular genotyping, a multiplex PCR was performed using the primers as reported previously (37). The PCR amplification consisted of an initial denaturation at 95°C for 10 min, followed by 35 cycles of denaturation at 90°C for 30 s, annealing at 58°C for 90 s, extension at 72°C for 90 s, and a final extension at 72°C for 10 min. Multiplex PCR-generated amplicons were further revalidated by singleplex PCR using the same primers. Strains positive for the capsular types were categorized as hypervirulent, and were analyzed for antibiotic susceptibility, AMR and virulence determinants, clonal relatedness, and *in vitro* virulence. Transmissibility of the carbapenemase gene was also assessed via conjugation. Genome-based and *in vivo* analyses were performed for strains harboring pLVPK-associated markers (*rmpA*, *rmpA2*, *iroN*, *iucA*, and *iutA*).

**Determination of MIC, and detection of antibiotic resistance and virulence determinants.** For hvKPs, carbapenem (ertapenem and meropenem) MICs were evaluated by broth microdilution (38), according to CLSI guidelines (35). Phenotypic detection of carbapenemases was also accomplished (Rosco Diagnostica A/S, Taastrup, Denmark).

PCR and sequencing were performed for the following AMR determinants: $\beta$-lactamases ($bla_{CTX-M,TEM,SHV,OXA}$), AmpCs ($bla_{MOX,CMY,DHA,ACC,MIR/ACT,FOX}$), carbapenemases ($bla_{KPC,SME,IMI,GES,NMC}$; $bla_{VIM,IMP,SPM,GIM,SIM,NDM}$; $bla_{OXA-48-like}$), aminoglycoside resistance genes (*rmt-A,B,C,D*, *armA*, and *aac(6′)-Ib*), and quinolone resistance genes (*qnr-A,B,C,D,S*, *qepA*, *oqx-A,B*, and *aac(6′)-Ib-cr*) (39, 40).

hvKP strains were tested for 15 different virulence-associated markers, including *fimH*, *mrkD*, *wabG*, *uge*, *wcaJ*, *magA*, *rmpA*, *rmpA2*, *entB*, *ybtS*, *iucA*, *iutA*, *iroN*, *allS*, and *kfuBC* by PCR (32, 39, 41). In this study, all PCRs were performed at least three times.

Significant differences in terms of AMR phenotypes, and carriage of resistance and virulence determinants between carbapenem-resistant and carbapenem-susceptible hvKPs were determined using $\chi^2$ or Fisher's exact test.

***In vitro* virulence assessment of the hvKP strains. (i) Biofilm-formation assay.** Biofilm-forming ability was assessed as described previously (42). Briefly, overnight cultures were adjusted to 0.5 McFarland, and 200 $\mu$L of bacterial suspensions were inoculated in 96-well plates. After incubation at 37°C for 24 h, biofilm was stained with 0.5% crystal violet, washed, and dissolved with 95% ethanol. Biofilm was quantified by measuring the absorbance at 595 nm ($OD_{595}$). Strong biofilm was considered when the readings were at least three times greater than the negative control.

**(ii) Serum resistance assay.** Serum bactericidal activity was determined as described previously (43). Briefly, 25 $\mu$L of $10^6$ CFU/mL bacterial inoculum was added to 75 $\mu$L of pooled normal human serum (1:3 vol/vol ratio) donated by healthy volunteers. The mixture was incubated at 37°C, and viable counts were taken at 0, 1, 2, and 3 h. Heat-inactivated serum was also prepared by heating at 56°C for 30 min.

For *in vitro* virulence assessment, hvKP-K1 strain SB42 ($magA^+$-$rmpA^+$-$rmpA2^+$-$iroN^+$-$iucA^+$-$iutA^+$) was taken as the positive control, and *K. pneumoniae* ATCC 700603 as the negative control.

**Pulsed-field gel electrophoresis (PFGE) and multi-locus sequence typing (MLST).** PFGE was employed to determine clonal relatedness using the PulseNet protocol (http://www.cdc.gov/pulsenet/protocols.htm) in a CHEF-DR III apparatus (Bio-Rad Laboratories, Hercules and CA). DNA digestion was carried out for 12 h at 37°C using XbaI enzyme. Digested DNA fragments were electrophoresed for 20 h at 14°C at a 120° included angle, linear ramp factor with switch times of 2.2 and 54.2 s at 6 V/cm. *Salmonella* serotype Braenderup H9812 was used as the size determining marker. The PFGE results were interpreted according to Tenover criteria (44). The Dice coefficient (1% tolerance and 1% optimization) and the unweighted pair-group method with the arithmetic mean (UPGMA) were used for the cluster analysis, using the FPQuest Software v4.5 (Bio-Rad Laboratories).

For sequence typing analyses, sequences of 7 conserved housekeeping genes of *K. pneumoniae* (*gapA*, *infB*, *mdh*, *pgi*, *phoE*, *rpoB*, and *tonB*) were submitted to the multilocus sequence typing (MLST) database (https://bigsdb.pasteur.fr/cgi-bin/bigsdb/bigsdb.pl?db=pubmlst__klebsiella_seqdef).

**Bacterial conjugation.** Conjugation experiments were performed for CR-hvKP strains by a solid mating assay at 37°C, using the *Escherichia coli* J53 (azide-resistant) recipient (39). Transconjugants (Tc) were selected on LB agar plates supplemented with 1 mg/L ertapenem and 100 mg/L sodium azide (Sigma-Aldrich, St. Louis, USA). Putative transconjugants were screened for AMR determinants (including $bla_{NDM}$), replicon types using PCR-based replicon typing (PBRT)-kit (Diatheva, Italy), and pLVPK-associated markers.

**DNA extraction, whole-genome sequencing, assembly, and annotation.** Based on the presence of pLVPK-associated markers (*rmpA*, *rmpA2*, *iroN*, *iucA*, and *iutA*), 2 NDM-1-producing CR-hvKPs, EN5180 (ST15-K54) and EN5289 (ST11-K2) strains were selected for WGS analysis. Genomic DNA (gDNA) of EN5180 and EN5289 were extracted using the QIAamp DNA minikit (Qiagen, Germany), according to the manufacturer's instructions. The purity and concentration of the extracted gDNA were fluorometrically determined using Qubit 3.0 (Thermo Fisher Scientific). The genome sequencing library was prepared using NEBNext Ultra II DNA Library Prep Kit following the manufacturer's instruction. Fragmented and adapter-tagged gDNA was amplified and sequenced on Illumina NextSeq 500 platform (Illumina Inc., San Diego, CA) with a read length of 151 bp. The sequence data was checked using FastQC v0.11.4, and adapters were trimmed using Cutadapt v2.5 (45, 46). The reads were *de novo* assembled using SPAdes v0.7.12-r1039 (47). Assessment of assembly metrics and genomic annotation were carried out using Quast v5.0.2 and Prokka v1.12 (48), respectively. Genomic islands were predicted using IslandViewer 4 (49).

**Analysis of the resistome, virulome, and other genetic elements.** Acquired AMR genes, plasmid replicons, and other elements in the studied genomes were determined using ResFinder (https://cge.cbs.dtu.dk/services/ResFinder/), PlasmidFinder (https://cge.food.dtu.dk/services/PlasmidFinder/), and the BIGSdb-Kp database (https://bigsdb.pasteur.fr/klebsiella/), respectively. Putative virulence factors were predicted using VRprofile (https://bioinfo-mml.sjtu.edu.cn/VRprofile/) with the BLASTp-based *Ha*-value > 0.64, which collected 2454 virulence determinants from the Virulence Factors Database (VFDB) (http://www.mgc.ac.cn/VFs/). Moreover, phage elements, CRISPR loci, and *in silico* K-antigen were identified using PHAST (http://phast.wishartlab.com/), CRISPRFinder (http://crispr.i2bc.paris-saclay.fr/Server/), and Kaptive (https://kaptive-web.erc.monash.edu/), respectively. The presence of pLVPK-like sequence in EN5180 and EN5289 was determined by progressiveMauve (http://darlinglab.org/mauve/user-guide/progressivemauve.html) using pLVPK (GenBank accession AY378100) as a reference.

**Comparative genomic and phylogenomic analysis.** For comparative genomics and phylogenomics, 11 publicly available reference genomes of hvKP [NTUH-2044 (GenBank accession NC_012731.1), 1084 (GenBank accession CP003785.1), ED23 (GenBank accession CP016814.1), ED2 (GenBank accession CP016813.1), RJF999 (GenBank accession CP014010.1), EN5275 (GenBank accession VINI00000000), RJF293 (GenBank accession CP014008.1), KCTC-2242 (GenBank accession CP002910.1), 1158 (GenBank accession CP006722.1), NUHL30457 (GenBank accession CP026586.1), and CG43 (GenBank accession CP006648.1)] were taken. Comparative BLAST+ and BLASTN analysis were carried out by Genome-to-Genome Distance Calculator (GGDC) v2.1 (http://ggdc.dsmz.de/) using the recommended formula 2 (identities/high-scoring pair lengths) and CGView server[BETA] (http://cgview.ca/), respectively. For phylogenomic analysis, we generated alignments by mapping query genomes and reference genomes using bowtie2 on the REALPHY v1.12 server (https://realphy.unibas.ch/realphy/). The phylogenetic tree was constructed by the Interactive Tree Of Life (iTOL) server (https://itol.embl.de/) using the REALPHY-generated .phy file.

***In silico* virulence plasmid analysis of EN5180 and EN5289.** From the draft genomes of EN5180 and EN5289, virulence plasmids were initially identified using MICRA pipeline (50). The annotation of the plasmids generated in MICRA were revalidated using RAST webserver (https://rast.nmpdr.org/). The mapping of plasmids were undertaken using Gview (https://server.gview.ca/), and pLVPK (GenBank accession AY378100) was used as a reference. Additionally, many of the protein-coding gene sequences present in plasmids were identified manually using blastx, and final maps were generated in Proksee server (https://proksee.ca/).

**Mouse lethality assay.** *In vivo* virulence of EN5180 and EN5289, carrying pLVPK-markers, was evaluated as described previously (51). Pathogen-free female 4 to 6-week-old BALB/c mice were infected with 100 $\mu$L of $1 \times 10^5$ CFU/mL bacterial suspensions through the intraperitoneal route. Symptoms and mortality were monitored every 24 h up to 7 days. *K. pneumoniae* ATCC 700603 was used as a low virulence control, and *K. pneumoniae* strain SB42 was taken as the hypervirulent control. Survival curves were evaluated using Kaplan-Meier analysis and the log-rank test. $P < 0.05$ was deemed significant. The animal experiments in this study were repeated three times.

**Data availability.** The raw sequencing reads of the CR-hvKP EN5180 and EN5289 strains were deposited in NCBI under BioProject number PRJNA684006. The genome sequences were submitted to the GenBank under the accession numbers JAELUV000000000 and JAELUW000000000 (Table 4).

## SUPPLEMENTAL MATERIAL

Supplemental material is available online only.
**SUPPLEMENTAL FILE 1**, PDF file, 1.3 MB.

## ACKNOWLEDGMENTS

We thank the clinicians and health care workers of the Department of Neonatology, IPGMER and SSKM Hospital for their support. We also thank Subhadeep De for his laboratory assistance. We extend our thanks to Dr Sylvain Brisse for providing the hvKP control strain SB42.

This study was supported by the Indian Council of Medical Research (ICMR) intramural fund. S. Mitra and S. Naha were supported by fellowships from ICMR.

The funding agency did not play any role in the study design, data collection, analysis and interpretation, writing of the report, or the decision to submit the work for publication.

We declare no conflicts of interest.

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
