## [Reviewer comments · Microbiology Spectrum]

Microbiology Spectrum

Hypervirulent *Klebsiella pneumoniae* causing neonatal bloodstream infections: emergence of NDM-1-producing hypervirulent ST11-K2 and ST15-K54 strains possessing pLVPK-associated markers

Subhankar Mukherjee, Punyasloke Bhadury, Shravani Mitra, Sharmi Naha, Bijan Saha, Shanta Dutta, and Sulagna Basu

Corresponding Author(s): Sulagna Basu, National Institute of Cholera and Enteric Diseases

Review Timeline:

Submission Date:	October 10, 2022
Editorial Decision:	November 28, 2022
Revision Received:	December 31, 2022
Accepted:	January 6, 2023

Editor: Justin Kaspar

Reviewer(s): The reviewers have opted to remain anonymous.

Transaction Report:

DOI: <https://doi.org/10.1128/spectrum.04121-22>

November 28, 2022

Dr. Sulagna Basu
ICMR-National Institute of Cholera and Enteric Diseases
Bacteriology
P33, CIT Road, Scheme XM, Beliaghata
Kolkata, West Bengal 700010
India

Re: Spectrum04121-22 (Hypervirulent *Klebsiella pneumoniae* causing neonatal bloodstream infections: emergence of NDM-1-producing hypervirulent ST11-K2 and ST15-K54 strains possessing pLVPK-associated markers)

Dear Dr. Sulagna Basu:

Link Not Available

Sincerely,

Justin Kaspar

Journals Department
Reviewer comments:

Reviewer #1 (Comments for the Author):

This is a very nice study of the emerging threat of hypervirulent *Klebsiella pneumoniae* in neonates. There are a few clarifications that the copy editors of the journal I think should be able to help you with. There are also comments on the methods, as well as some requests for clarification in the conclusion and tables.

Reviewer #2 (Comments for the Author):

Here authors performed antibiotic susceptibility, molecular characterization, evaluation of clonality, in vitro virulence, and

transmissibility of carbapenemase genes for neonatal septicemic hvKP/CR-hvKP strains. About one-fourth (26%, 28/107) of the studied strains were categorized as hvKP, and 32.1% (9/28) strains were categorized as CR-hvKP. In addition, WGS and mouse lethality assay were performed on strains harboring pLVPK associated markers. This study reported the NDM-1-producing hvKP ST11-K2 and ST15-K54 strains causing fatal neonatal sepsis, and demonstrated the presence of pLVPK-associated markers and blaNDM-1 in high-risk clones and the co-transmission of these genes via conjugation.

Overall, this manuscript is well written and of interest to researchers in the related fields. Here are some specific points that need to be addressed.

Line 34-36: "About one-fourth (26%, 28/107) of the studied strains" This study mainly focused on 28 hvKP strains, especially CR-hvKP strains, and 107 strains mentioned here can be confusing.

Line 120-124: Provide detailed description on the origin of bacterial strains, how many patients participated in the study, and how the strains were isolated from blood.

Line 127-129: Provide a detailed description and reference for this method.

Line 297 and Table 2: It is better to provide definitions for Inborn and Outborn.

Staff Comments:

Preparing Revision Guidelines

Please return the manuscript within 60 days; if you cannot complete the modification within this time period, please contact me. If you do not wish to modify the manuscript and prefer to submit it to another journal, please notify me of your decision immediately so that the manuscript may be formally withdrawn from consideration by Microbiology Spectrum.

**Hypervirulent *Klebsiella pneumoniae* causing neonatal bloodstream infections:**
**emergence of NDM-1-producing hypervirulent ST11-K2 and ST15-K54 strains**
**possessing pLVPK-associated markers**

**Running title:** CR-hvKP ST11-K2 and ST15-K54 causing fatal neonatal sepsis.

Subhankar Mukherjee^{a*}, Punyasloke Bhadury^b, Shravani Mitra^a, Sharmi Naha^a, Bijan Saha^c,
Shanta Dutta^a, Sulagna Basu^{a#}

**Affiliations:**

8 ^a *Division of Bacteriology, ICMR-National Institute of Cholera and Enteric Diseases.*
*Kolkata, West Bengal, India.*

10 ^b *Integrative Taxonomy and Microbial Ecology Research Group, Department of Biological*
*Sciences, Indian Institute of Science Education and Research Kolkata, Mohanpur, Nadia,*
*West Bengal, India.*

13 ^c *Department of Neonatology, Institute of Postgraduate Medical Education & Research,*
*SSKM Hospital, Kolkata, West Bengal, India.*

* **Present address:** *Department of Zoology, Government General Degree College, Singur,*
*Hooghly, West Bengal, India.*

**Keywords:** *Klebsiella pneumoniae*, neonatal sepsis, antibiotic resistance, carbapenem
resistance, hypervirulence, India.

**# Corresponding author:**

Sulagna Basu, Division of Bacteriology, ICMR-National Institute of Cholera and Enteric
Diseases, P-33, C.I.T. Road, Scheme XM, Beliaghata, Kolkata-700 010, West Bengal, India.

E-mail address: supabasu@yahoo.co.in; basus.niced@gov.in

Telephone: +91-33-2353 7469/7470, 23705533/4478/0448; ext: 3055. Fax: +91-33-2363,
2370 5066

[revised manuscript text omitted]

antimicrobials (**Supplementary Figure 1**). Twenty-nine hvKP strains were detected, of
them, 28 hvKPs are characterized here, as ly on CR-hvKP strain EN5275 (ST23-K1) has
already been published (34).

Most of the recovered hvKPs were resistant to cephalosporin, cephamycin, monobactam,
aminoglycosides, fluoroquinolone, and trimethoprim/sulfamethoxazole (**Table 1**). Among the
studied hvKPs, only 32.1% (9/28) strains were detected as metallo- β -lactamase producers,
resistant to carbapenems (MICs ranged from 8 to 64 mg/L), and were categorized as CR-
hvKP.

AMR determinants

Various β -lactamase genes (ESBLs, AmpCs, and carbapenemases), aminoglycoside
resistance genes, and quinolone resistance genes were identified in hvKPs (**Table 1**).

Moreover, out of nine CR-hvKPs, seven harbored *bla*_{NDM-1} and two harbored *bla*_{NDM-5}. One
strain co-harbored *bla*_{NDM-5} and *bla*_{OXA-232}. Among the several AMR determinants, *rmtC* ($P =$
0.048), *aac(6')-ib* ($P = 0.017$), and *qnrS1* ($P = 0.023$) were significantly higher in CR-hvKPs
(**Table 1**). **Figure 1** depicts the distribution of AMR determinants.

Virulence-associated analysis

K2 (64.2%, 18/28) was the most prevalent of the identified capsular types, followed by K54
(17.8%, 5/28) and K57 (14.3%, 4/28) (**Table 1**). Moreover, K2 and K54 were detected in
both carbapenem-resistant and carbapenem-susceptible strains. Several virulence
determinants, viz., *wabG*, *kfuBC*, *mrkD*, *fimH*, *entB*, *uge*, *ybtS*, and *wcaJ* were identified in
hvKPs, of them, *wcaJ* ($P < 0.001$) and *ybtS* ($P = 0.010$) were significantly higher in CR-
hvKPs (**Table 1**). In contrast, pLVPK-associated markers (*rmpA*, *rmpA2*, *iucA*, *iutA*, and

*iroN*) were detected only in two NDM-1-producing CR-hvKP strains (EN5180 and EN5289).

**Figure 1** depicts the distribution of various virulence factors.

Additionally, 42.8% (12/28) of strains possessed strong biofilm-forming ability ($OD_{595} =$
0.82–1.23) compared to the negative control ($OD_{595} = 0.27$), and 78.5% (22/28) of strains
exhibited high level of serum resistance (100% viable counts after 3 h of incubation)
compared to the positive control. **Supplementary Table 1** summarizes *in vitro* virulence
results.

**Clonal relatedness among hvKP strains**

PFGE revealed that most of the studied hvKPs (60.7%, 17/28) were clonally diverse.
However, among the nine CR-hvKPs, EN5327 and EN5329 were clonal and among the
nineteen carbapenem-susceptible strains, three distinct clonal-clusters were observed (Cluster
I – EN5230, EN5233, EN5235, and EN5236; Cluster II – EN5303, EN5306, and EN5323;
Cluster III – EN5291 and EN5292), while rest of the others appeared to be distinct (**Figure**
**1**).

The hvKPs consisted of 12 different STs (ST11, ST14, ST15, ST29, ST65, ST268, ST307,
ST520, ST711, ST1035, ST1308, ST2343) (**Figure 1**). Of them, ST14 was the most common
(32.1%, 9/28) and was associated with both carbapenem-resistant and carbapenem-
susceptible populations. Additionally, ST11, ST15, ST65, ST268, and ST307 were associated
only with CR-hvKPs. Other STs (ST29, ST520, ST711, ST1035, ST1308, and ST2343) were
detected only in carbapenem-susceptible strains.

**Clinical outcome of the neonates infected with hvKP strains**

**Table 2** represents the clinical status of the hvKP-infected neonates. Out of 28 neonates,
sixteen (57.1%) were **5** born, majority were delivered preterm (53.5%, 15/28), and were of
low-birth-weight (57.1%, 16/28) or extremely-low-birth-weight (17.8%, 5/28). ~~About~~ 39.2%
(11/28) of neonates who succumbed to the condition, suffered from severe clinical

complications, as listed in **Table 2**. Notably, hvKP-K2 strains were found responsible for
33.3% (6/18) of neonatal mortality.

**Analysis of transconjugants**

[revised manuscript text omitted]

***In vivo* virulence of EN5180 and EN5289**

A significant difference in mouse lethality was noticed between the CR-hvKP strain EN5289
and the other reference strains. The Kaplan-Meier survival analysis revealed that mice
infected with EN5289 portrayed virulence similar to that of the hvKP-K1 positive control
strain SB42 (**Figure 4**), indicating that EN5289 was equivalently virulent to the well-
established hvKP-K1 strain in this model. Whereas, despite harboring all the hypervirulent
genetic traits, higher inoculum (10^5 CFUs) of EN5180 did not lead to significant mouse
lethality, demonstrating the moderate *in vivo* virulence of EN5180.

**Discussion**

The recent emergence of hvKP in the clinical context is a serious public-health concern,
primarily because besides causing severe life-threatening infections, these strains are also
gradually becoming drug-resistant (**14**). Additionally, the detection of hvKP and its etiology
are not yet well understood due to the absence of a universally agreed consensus
definition/marker for hypervirulence and relied mostly on genotypic methods. Hence, clinical
microbiology laboratories are sometimes unable to differentiate hvKP from cKP in routine
diagnosis, affecting the overall treatment regimen severely (**35, 36**).

[revised manuscript text omitted]

- 498 7. Zhang X, Li X, Wang M, Yue H, Li P, Liu Y, Cao W, Yao D, Liu L, Zhou X, Zheng R, Bo
499 T. 2015. Outbreak of NDM-1-producing *Klebsiella pneumoniae* causing neonatal infection in
a teaching hospital in mainland China. *Antimicrob Agents Chemother* 59:4349–51.
- 8. WHO. 2017. Global Priority List of Antibiotic-resistant Bacteria to Guide Research,
Discovery, and Development of New Antibiotics.
[https://www.who.int/medicines/publications/global-priority-list-antibiotic-resistant-](https://www.who.int/medicines/publications/global-priority-list-antibiotic-resistant-bacteria/en/)
[bacteria/en/](https://www.who.int/medicines/publications/global-priority-list-antibiotic-resistant-bacteria/en/).
- 9. Centers for Disease Control and Prevention U.S. 2013. Antibiotic resistance threats in the
United States. <https://stacks.cdc.gov/view/cdc/20705>.
- 10. Shon AS, Bajwa RP, Russo TA. 2013. Hypervirulent (hypermucoviscous) *Klebsiella*
*pneumoniae*: a new and dangerous breed. *Virulence* 4:107-18.
- 11. Paczosa MK, Mecsas J. 2016. *Klebsiella pneumoniae*: going on the offense with a strong
defense. *Microbiology and Molecular Biology Reviews* 80:629-61.
- 12. Russo TA, Marr CM. 2019. Hypervirulent *Klebsiella pneumoniae*. *Clinical microbiology*
*reviews* 32:e00001-19.

[revised manuscript text omitted]

ST, sequence type; M, male; F, female; LUCS, lower uterine segment cesarean section; NVD, normal vaginal delivery; HIE-II, hypoxic-ischemic encephalopathy grade II;

ELBW, extremely low-birth-weight (<1000 g); IVH-III, intraventricular hemorrhage grade III; MSL, meconium-stained liquor; BCA, bilateral choanal atresia; IUGR, intrauterine

growth restriction; DIC, disseminated intravascular coagulation; LBW, low-birth-weight (<2500 g); PR bleeding, per rectal bleeding; Preterm, <37 weeks of gestational age;

GDM, gestational diabetes mellitus.

**Table 3.** Microbiological characteristics of the *bla*_{NDM}-harboring parental strains and their transconjugants.

Strains	ST	MIC (mg/L)		AMR determinants	pLVPK-associated markers	Inc group
		ETP	MEM			
EN5180	15	16	32	bla _{NDM-1} , bla _{CTX-M-15} , bla _{TEM-1} , bla _{SHV} , bla _{OXA} , bla _{CMY-6} , armA , rmtC , oqxAB , aac (6')- ib , aac (6')- ib-cr	rmpA , rmpA2 , iucA , iutA , iroN	A/C, FIIK, HIB-M, FIB-M
Tc-EN5180	-	4	8	bla _{NDM-1} , bla _{TEM-1} , rmtC , aac (6')- ib	-	A/C
EN5187	307	32	64	bla _{NDM-1} , bla _{CTX-M-15} , bla _{SHV} , bla _{OXA} , qnrB1 , qnrS1 , oqxAB , aac (6')- ib , aac (6')- ib-cr	-	FIIK, L/M, FII
Tc-EN5187	-	8	8	bla _{NDM-1} , bla _{CTX-M-15} , qnrB1 , qnrS1 , aac (6')- ib , aac (6')- ib-cr	-	FIIK
EN5199	14	>32	>32	bla _{NDM-5} , bla _{OXA-232} , bla _{CTX-M-15} , bla _{TEM-1} , bla _{SHV} , bla _{OXA} , rmtB , oqxAB , aac (6')- ib , aac (6')- ib-cr	-	R, FIIK, FII, FIB, FIA, Col
Tc-EN5199	-	32	24	bla _{NDM-5} , bla _{OXA-232} , bla _{CTX-M-15} , bla _{TEM-1} , rmtB	-	R, FIIK, Col
EN5206	14	32	32	bla _{NDM-1} , bla _{CTX-M-15} , bla _{TEM-1} , bla _{SHV} , qnrB1 , qnrS1 , oqxAB , aac (6')- ib	-	FII, FIIK, R
Tc-EN5206	-	8	4	bla _{NDM-1} , bla _{CTX-M-15} , qnrS1 , aac (6')- ib	-	FIIK
EN5289	11	64	32	bla _{NDM-1} , bla _{CTX-M-15} , bla _{TEM-1} , bla _{SHV} , bla _{OXA} , bla _{DHA-1} , armA , qnrS1 , oqxAB , aac (6')- ib , aac (6')- ib-cr	rmpA , rmpA2 , iucA , iutA , iroN	FIIK, HIB-M, FIB-M, HI1
Tc-EN5289	-	8	4	bla _{NDM-1} , bla _{CTX-M-15}	rmpA , rmpA2 , iucA , iutA , iroN	HIB-M, FIB-M
EN5298	15	32	64	bla _{NDM-5} , bla _{CTX-M-15} , bla _{TEM-1} , bla _{SHV} , bla _{OXA} , qnrB1 , qnrS1 , oqxAB , aac (6')- ib , aac (6')- ib-cr	-	FIIK, FIB, FIA
Tc-EN5298	-	4	2	bla _{NDM-5} , bla _{CTX-M-15} , qnrS1 , aac (6')- ib , aac (6')- ib-cr	-	FIIK
EN5327	65	16	8	bla _{NDM-1} , bla _{CTX-M-15} , bla _{TEM-1} , bla _{OXA} , bla _{ACT-69} , rmtC , qnrB1 , qnrS1 , oqxAB , aac (6')- ib , aac (6')- ib-cr	-	FII, FIIK, X3
Tc-EN5327	-	8	4	bla _{NDM-1} , bla _{CTX-M-15} , bla _{TEM-1} , rmtC , qnrB1 , aac (6')- ib , aac (6')- ib-cr	-	FII
EN5329	65	16	8	bla _{NDM-1} , bla _{CTX-M-15} , bla _{TEM-1} , bla _{SHV} , bla _{OXA} , bla _{ACT-69} , rmtC , qnrB1 , qnrS1 , oqxAB , aac (6')- ib , aac (6')-	-	FII, FIIK, X3

				ib-cr		
Tc-EN5329	-	8	4	bla _{NDM-1} , bla _{CTX-M-15} , bla _{TEM-1} , rmtC , qnrB1 , aac(6') - ib , aac(6') - ib-cr	-	FII
EN5337	268	16	8	bla _{NDM-1} , bla _{CTX-M-15} , bla _{SHV} , bla _{DHA-1} , qnrB1 , qnrS1 , oqxAB , aac(6') - ib	-	R, FIIK, HIB- M, FIB-M
Tc-EN5337	-	4	2	bla _{NDM-1} , bla _{CTX-M-15} , qnrB1 , qnrS1 , aac(6') - ib	-	R, FIIK

ST, sequence type; ETP, ertapenem; MEM, meropenem; pLVPK, large virulence plasmid of *Klebsiella*; Tc, *E. coli*

J53 transconjugants selected in LB agar medium containing 1 mg/L ertapenem and 100 mg/L sodium azide.

Strains with bold formatting signifies pLVPK-markers-harboring CR-hvKP strains and their respective Tc.

Table 4. Genome-level characteristics of neonatal septicemic CR-hvKP strains EN5180 and EN5289.

Variable	Strain information	
	K. pneumoniae EN5180	K. pneumoniae EN5289
Strain name	K. pneumoniae EN5180	K. pneumoniae EN5289
Sequence type	ST15	ST11
Capsular type	K54	K2
Genomic features		
Genome size (bp)	6131075	5967685
Contig numbers	115	107
GC content (%)	56.43	56.68
CDS	5974	5902
rRNA genes	14	34
tRNA genes	74	80
tmRNA genes	1	1
ncRNA	13	11
Pseudogenes	146	135
Repeat region	3	1
Antibiotic resistance genes		
Aminoglycoside resistance	aac(6)-Ib3, aadA1, armA, rmtC	aac(6)-Ib, aadA1, armA
β -lactam resistance	bla_{SHV-28}, bla_{SHV-106}, bla_{TEM-1A}, bla_{OXA-9}, bla_{CTX-M-15}, bla_{CMY-6}, bla_{NDM-1}	bla_{SHV-182}, bla_{TEM-1B}, bla_{OXA-9}, bla_{CTX-M-15}, bla_{DHA-1}, bla_{LAP-2}, bla_{NDM-1}
Quinolone resistance	aac(6)-Ib-cr, oqxA, oqxB	aac(6)-Ib-cr, oqxA, oqxB, qnrS1
Fosfomycin resistance	fosA	fosA
Macrolide resistance	msr(E)	msr(E)
Sulphonamide resistance	sul1	sul1
Trimethoprim resistance	-	tet(A)
Plasmid types		

Col plasmids	Col440I	Col440I, Col(BS512)
F plasmids	IncFIB(pKPHS1), IncFIB(pNDM-Mar), IncFIB(pQil), IncFII(K)	IncFIB(pNDM-Mar), IncFIB(pQil), IncFII(K)
Other plasmids	IncHI1B(pNDM-MAR), IncA/C2	IncHI1B(pNDM-MAR)
Heavy metal resistance genes		
Silver resistance	silCERS	silCERS
Tellurite resistance	terABCDEWXYZ	terABCDEWXYZ
Efflux pumps and regulators		
RND family efflux pump	acrAB	acrAB, mexB, mdtL
HTH type transcriptional regulator	acrR	acrR
AraC/XyIS family transcriptional activator	marAB, soxRS, rob, ramA, rarA	marAB, soxRS, rob, ramA, rarA
TetR family transcriptional regulator	ramR	ramR
HTH LuxR type transcriptional regulator	sdiA	sdiA
rRNA transcriptional activator	fis	fis
Suppressor of efflux pump	envR	envR
Quinolone and Olaquinox efflux pump	oqxABR	oqxABR
Virulence genes		
Type 3 fimbriae	mrkABCFHIJ	mrkABCFHIJ
Type 1 fimbriae	fimABCDEFGHIK	fimABCDEFGHIK
Type IV pili biosynthesis	-	pilU
ABC iron transporter	kfuABC	kfuABC
Aerobactin	iucABCD; iutA	iucABCD; iutA
Enterobactin	entABCDEFS; fepABCDG; fes	entABCDEFS; fepABCDG; fes
Salmochelins	iroBCDEN	iroBCDEN
Yersiniabactin	fyuA; irp1, irp2; ybtAEPQSTUX	fyuA; irp1, irp2; ybtAEPQSTUX
Allantoin utilization	allABCDRS	allABCDRS

Capsule regulators	rcaAB; rmpA, rmpA2	rcaAB; rmpA, rmpA2
pLVPK-derived loci	rmpA, rmpA2; iroBCDEN; iucABCDiutA; peg-344	rmpA, rmpA2; iroBCDEN; iucABCDiutA; peg-344
Secretion system	T6SS (I - III)	T6SS (I - III)
Other factors (Autotransporter, Iron/ manganese uptake, and Stress adaptation)	flu; sitABCD; mntB	-
BioProject number		PRJNA684006
Accession number	JAELUV000000000	JAELUW000000000

CDS, coding sequence; tmRNA, transfer-messenger RNA; ncRNA, non-coding RNA; RND, resistance-
nodulation-division; HTH, helix-turn-helix.

**Figure 1a**

**Figure 2**

**Figure 3a**

**Figure 3b**

Figure 4

Response to Reviewers

Point-by-point responses to the issues raised by the reviewers

(All corrections and/or changes in the “Marked-Up Manuscript” are highlighted in yellow).

1. As per the reviewer’s suggestion, the word (**immensely**) has been deleted from **line no. 29**.

2. As suggested, the word (**evaluated**) has been incorporated in **line no. 33**.

3. The correction has been incorporated (**The majority**) in **line no. 54**.

4. The correction has been made according to the reviewer’s suggestion and has been included (**infecting syndromes**) in **line no. 78**.

5. The necessary change has been made in **line no. 88**.

6. Helpful to reader to understand that hypervirulence genes are not always the same as hypermucoviscous genes See Walker, Kimberly et al PMID: 32062153 as well as: Catalan-Najarra et al: 28402698.

Answer: As suggested, the clarification of hypervirulence and hypermucoviscosity has been provided in **line no. 95-97**. The reference has been included both in **line no. 97** and **522-524**.

7. Change has been included (**The majority**) in **line no. 98**.

8. As suggested, the sentence (**in order to better understand their genomes**) has been deleted in **line no. 113**.

9. What does the level III mean?

Answer: The meaning of level III unit (**Neonatal intensive care unit**) has now been included in the revised manuscript in **line no. 124**.

10. Species confirmation via VITEK-2 sometimes misses variicola. Would comment on confirmation via WGS.

Answer: This is an important comment. Because of the complexities of the *K. pneumoniae* complex, several researchers proposed different methods of distinguishing *K. variicola* from *K. pneumoniae*. However, phylogenetic analysis of 16S rRNA and *rpoB* genes are most commonly used to properly identify *K. variicola*. In the present study, we have performed WGS only for two CR-hvKP strains and those were *K. pneumoniae*. Other strains were identified by automated VITEK-2 compact system and strain characterization/discrimination was further validated by sequence typing analysis via targeting the house-keeping genes of *K. pneumoniae* (which are different in *K. variicola*). Hence though VITEK was used, the identified STs of the strains reliably confirmed them to be *K. pneumoniae*.

11. How exactly were they screened? Methodology.

Answer: The methodology of capsular genotyping has been included as per the reviewer's suggestion (**line no. 137-142**).

12. How was this assessed?

Answer: Transmissibility of the carbapenemase gene was accomplished by **conjugation (line no. 145)** and the methodology has already been provided in **line no. 194-200**.

13. You did PCR and sequencing for all of these?

Answer: PCR was performed for all antimicrobial resistance (AMR) determinants mentioned in the manuscript. In addition, Sanger sequencing was carried out for the identified AMR determinants, such as ESBL (*bla_{CTX-M}*), plasmid-mediated AmpCs (*bla_{CMY}*, *bla_{DHA}*, and *bla_{ACT}*), carbapenemases (*bla_{NDM}* and *bla_{OXA-48-like}*), and plasmid-mediated quinolone resistance genes (*qnrB* and *qnrS*).

14. Needs an article here "a study" or "previous work". Also not sure why you wouldn't include this in the entire analysis

Answer: As per the reviewer's suggestion the word **previous work** has been included in **line no. 270**.

15. Clarify what outborn means

Answer: The clarification of 'outborn' has been incorporated both in **line no. 312-313** and in **the footnote of Table 2**.

16. As per the reviewer's suggestion, the word (**about**) has been deleted from **line no. 314**.

17. I think you can just reference the table rather than repeat the findings here.

Answer: The modifications have been made in **line no. 340-341**.

18. Really comment needs to be restricted to this mouse model. Unclear if this would be replicated.

Answer: In the present study, the animal experiments were repeated thrice. The clarification and/or medications have been made in **line no. 259-260, 434**. The sentence (**demonstrating the moderate *in vivo* virulence of EN5180**) has now been removed from **line no 376**.

19. Revise this sentence.

Answer: The necessary modifications have been included in **line no. 378-380**.

20. The correction has been made according to the reviewer's suggestion and the sentence "**potentially affecting the treatment regimen**" has been incorporated in **line no. 381**.

21. This sentence needs to be revised

Answer: The sentence has now been revised in **line no. 389-391**.

22. The correction has been made as per the reviewer's suggestion and the word **therefore** has been deleted from **line no. 398**.

23. You can speculate this but you haven't shown this in this paper. Would need more evidence.

Answer: According to the reviewer's suggestion, the correction has been made in **line no. 431** and the reference has been incorporated in **line No. 433** and **634-637**.

24. As suggested, the correction (**The** mouse lethality assay) has been incorporated in **line no. 434**.

25. The correction has been made as per the reviewer's suggestion and the article (**the**) has been deleted in **line no. 445**.

26. The reference has been moved to end of the sentence in **line no. 452**.

27. The necessary changes have been made in **line no. 458-459**.

28. Again not always the same as hypervirulence.

Answer: The clarification has already been provided in **line no. 95-97**.

29. Clarify the AMR phenotypes portion of the table. Assuming here that the CR-hvKP column is indicating number of strains resistant but it needs to be spelled out more clearly.

Answer: Clarification of the AMR phenotypes portion has been incorporated in Table 1

30. Copy editing clarification

Answer: The correction has been made (in Table 1) as per the reviewer's suggestion.

Reviewer comments:

Reviewer #1 (Comments for the Author):

This is a very nice study of the emerging threat of hypervirulent *Klebsiella pneumoniae* in neonates. There are a few clarifications that the copy editors of the journal I think should be able to help you with. There are also comments on the methods, as well as some requests for clarification in the conclusion and tables.

Answer: Thank you very much for your consideration. All comments and clarifications have been addressed as per the reviewer's suggestion.

Reviewer #2 (Comments for the Author):

Here authors performed antibiotic susceptibility, molecular characterization, evaluation of clonality, in vitro virulence, and transmissibility of carbapenemase genes for neonatal septicemic hvKP/CR-hvKP strains. About one-fourth (26%, 28/107) of the studied strains were categorized as hvKP, and 32.1% (9/28) strains were categorized as CR-hvKP. In addition, WGS and mouse lethality assay were performed on strains harboring pLVPK associated markers. This study reported the NDM-1-producing hvKP ST11-K2 and ST15-K54 strains causing fatal neonatal sepsis, and demonstrated the presence of pLVPK-associated markers and blaNDM-1 in high-risk clones and the co-transmission of these genes via conjugation.

Overall, this manuscript is well written and of interest to researchers in the related fields. Here are some specific points that need to be addressed.

Line 34-36: "About one-fourth (26%, 28/107) of the studied strains" This study mainly focused on 28 hvKP strains, especially CR-hvKP strains, and 107 strains mentioned here can be confusing.

Answer: 107 *K. pneumoniae* were isolated from the blood of neonates during the study period. Given the importance of estimating the burden of hvKP among all neonatal cases with *K. pneumoniae* infection, this has been mentioned. The reader gets an estimate of the burden of hvKP which would otherwise not be reflected.

Line 120-124: Provide detailed description on the origin of bacterial strains, how many patients participated in the study, and how the strains were isolated from blood.

Answer: The clarification has been provided in **line no. 122**. In addition, the detailed description of blood culture has been included in **line no. 125-133**.

Line 127-129: Provide a detailed description and reference for this method.

Answer: The methodology of capsular genotyping and the reference have been included as per the reviewer's suggestion in **line no. 137-142**.

Line 297 and Table 2: It is better to provide definitions for Inborn and Outborn.

Answer: As suggested, the necessary changes have been incorporated both in **line no. 312-313** and in the **footnote of Table 2**.

January 6, 2023

Dr. Sulagna Basu
National Institute of Cholera and Enteric Diseases
Bacteriology
P33, CIT Road, Scheme XM, Beliaghata
Kolkata, West Bengal 700010
India

Re: Spectrum04121-22R1 (Hypervirulent *Klebsiella pneumoniae* causing neonatal bloodstream infections: emergence of NDM-1-producing hypervirulent ST11-K2 and ST15-K54 strains possessing pLVPK-associated markers)

Dear Dr. Sulagna Basu:

Your manuscript has been accepted, and I am forwarding it to the ASM Journals Department for publication. You will be notified when your proofs are ready to be viewed.

Sincerely,

Justin Kaspar
Editor, Microbiology Spectrum